# Multicentennial cycles in continental demography synchronous with solar activity and climate stability

Kai W. Wirtz [1,2] ✉, Nicolas Antunes [3,4], Aleksandr Diachenko [5], Julian Laabs [6,7], Carsten Lemmen [1], Gerrit Lohmann [8], Rowan McLaughlin [9], Eduardo Zorita[1] & Detlef Gronenborn [3,10]

Human population dynamics and their drivers are not well understood, especially over the long term and on large scales. Here, we estimate demographic growth trajectories from 9 to 3 ka BP across the entire globe by employing summed probability distributions of radiocarbon dates. Our reconstruction reveals multicentennial growth cycles on all six inhabited continents, which exhibited matching dominant frequencies and phase relations. These growth oscillations were often also synchronised with multicentennial variations in solar activity. The growth cycle for Europe, reconstructed based on >91,000 radiocarbon dates, was backed by archaeology-derived settlement data and showed only a weak correlation with mean climate states, but a strong correlation with the stability of these states. We therefore suggest a link between multicentennial variations in solar activity and climate stability. This stability provided more favourable conditions for human subsistence success, and seems to have induced synchrony between regional growth cycles worldwide.

Common reconstructions of population dynamics on continental and global scales for the last millennia leave the impression that absolute numbers had been growing rather continuously over time[1,2]. However, when focusing on past local to regional scales, the archaeological record discloses a much more disruptive picture. Summed probability distributions (SPDs) of archaeological radiocarbon ([14]C) dates are increasingly used to reconstruct dynamically rich demographic trajectories[3–5]. The drivers of these dynamics are poorly understood. Potential causes for discontinuities in human demographic and social development can be exogenous factors such as climate anomalies, endogenous processes such as war, migration, and disease, or combinations of exogenous and endogenous processes[6–9]. Often, exogenous

factors were suggested as critical trigger since collapses of past societies sometimes coincided with events recorded in Holocene climate proxies[10–15]. However, contradicting the role of large-scale climate as a demographic driver, regional changes in SPDs within, e.g., Europe[3], South America[4], or Southwest Asia[5] appeared as poorly synchronized, or average out at a continental scale[16]. A rather small exogenous control seemed to be confirmed by the first more systematic assessment of the climate–society relationship by Freeman et al., who used statistical measures to reveal a moderate synchrony between regional SPDs at multicentennial scales[6]. This synchrony decreased with increasing distance of regions and was unrelated to solar forcing as a simple representative of global climate variability. Therefore, Freeman et al.

[1]Helmholtz Zentrum Hereon, 21501 Geesthacht, Germany. [2]Kiel University, 24118 Kiel, Germany. [3]Leibniz-Zentrum für Archäologie, 55116 Mainz, Germany. [4]German Archaeological Institute, 14195 Berlin, Germany. [5]Institute of Archeology of the National Academy of Sciences, 04210 Kyiv, Ukraine. [6]CRC 1266 Scales of Transformation, Kiel University, 24118 Kiel, Germany. [7]Department of History, University of Leipzig, 04109 Leipzig, Germany. [8]Alfred Wegener Institute, Helmholtz Centre for Polar and Marine Research, 27515 Bremerhaven, Germany. [9]Hamilton Institute, Maynooth University, Co Kildare, Ireland. [10]Johannes Gutenberg-University, 55116 Mainz, Germany. ✉e-mail: kai.wirtz@hereon.de

concluded that predominantly endogenous processes were responsible for regional synchrony.

Here, we reconstruct human population dynamics for all six inhabited continents 9–3 ka BP and identify their relation to climate variability on different spatial scales. Climate variability is estimated based on a broad compilation of a hundred paleoclimatic proxy time-series, while the relative explanatory power of climatic and endogenous processes is evaluated in a systematic approach comprising statistical modeling and different similarity measures. Our study employed SPDs of 178,833 [14]C-dated archaeological samples worldwide (Fig. 1), from which all non-European [14]C-dates originated from the p3k14C-database[17], while the majority of the 91,173 European dates derived from our compilation of sources specified in Tab. S1. Nearly all studies contrasting paleoclimate with paleodemography refer to extrema in SPDs, whereas we calculated a relative growth rate (RGR, see Methods). This allows a more precise causal inference analysis since drivers act on growth rate rather than on population density. For example, during a 'boom' defined as a peak in population size, the latter already starts to decline, which describes a crisis situation. Therefore, we here distinguish (growth) booms and busts according to the sign of RGR.

## Results and Discussion

### Multicentennial continental growth cycles

In all six continents, our reconstruction reveals a persistent sequence of phases with positive and negative RGR (Fig. 2a–d). The typical duration of a combined boom-bust phase from about four to eight centuries is best visible in the frequency domain. The spectral intensity of continental RGRs peaks at or near $(360\ a)^{-1}$, $(500\ a)^{-1}$, and $(680\ a)^{-1}$ (Fig. 3a–b), which is most apparent for Europe. In other continents, also a faster and a slower spectral mode appear at $(260\ a)^{-1}$ and $(950\ a)^{-1}$. Not only the dominant multicentennial periodicities, but also the phases of the RGR cycles match to a high degree: phase overlap and trim correlation $r_T$ (defined in the Methods) between individual continental cycles can be well beyond 60% and $r_T = 0.5$, respectively (Fig. 2a–d), and even reach 80% and $r_T = 0.8$ between the European and South American growth cycles (Fig. 2d). Apart from the period 6.5 to 5.5 ka BP, the demographic trajectories of Europe and South America show a high level of agreement.

### Synchrony of growth with solar forcing

The match of continental growth cycles in terms of dominant frequencies and phase may be rooted in a common external driver.

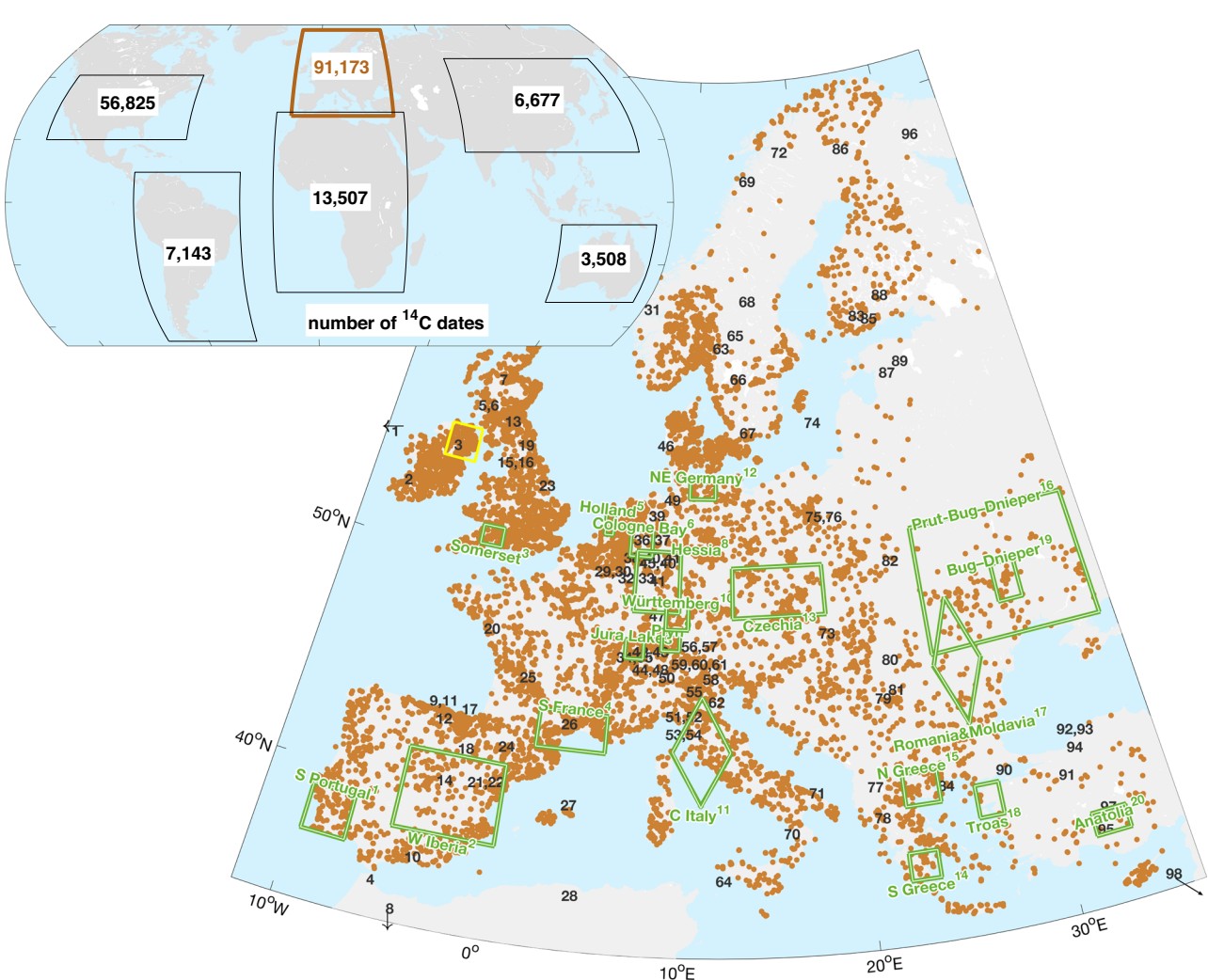

**Fig. 1 | Global partitioning of [14]C-dates into continental boxes.** The sometimes smaller extension of the boxes compared to the geographical boundaries of a continent reflect wide gaps in [14]C-dated sites such as in South Asia. For Europe and western Anatolia, the focal study area, 15,089 archeological sites with 91,173 dates are shown as small brown circles. Subregions with independent reports of occupation density are approximated by green quadrangles (Tab. S4). Numbers indicate the locations of the 98 paleoclimate proxy sites (Tab. S2), and the yellow box the distribution of Northern Irish bogs used for the annual tree-ring reconstruction. Capital letters `NSWE' indicate the four cardinal directions.

Although Holocene climate variability differed between world regions[18], an underlying forcing may follow from the varying activity of the sun: total solar irradiance (TSI) has been suggested as a proxy or even driver for regional (paleo)climate variability[19,20], further detailed in Sec. S2. Indeed, variations in TSI exhibit a cyclicity very similar to the one of continental growth trajectories, as quantified by very high

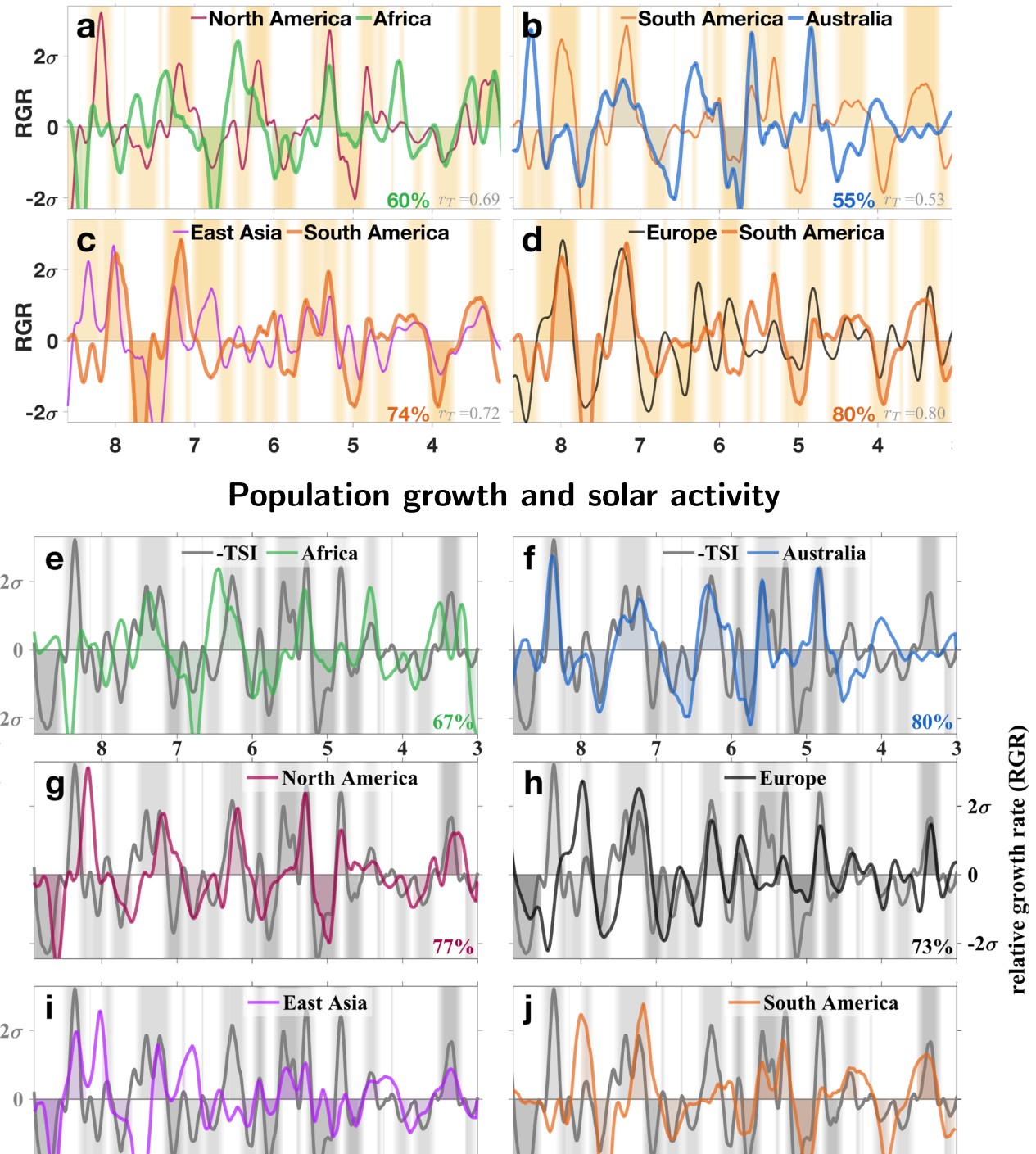

**Fig. 2 | Fluctuations in relative growth rate (RGR, change in Summed Probability Distribution, SPD) in all populated continents. a–d** Phase relationships between continental population cycles. Phases of either positive or negative RGR of the first continent are shaded. Percentage shows relative phase overlaps (see Methods). **d–j** Phase relationships between continental RGR with negative total solar irradiance (TSI, with shading for positive and negative phases, resp.). All time-series in (**a–j**) were normalized (division by standard deviation $\sigma$), smoothed and detrended.

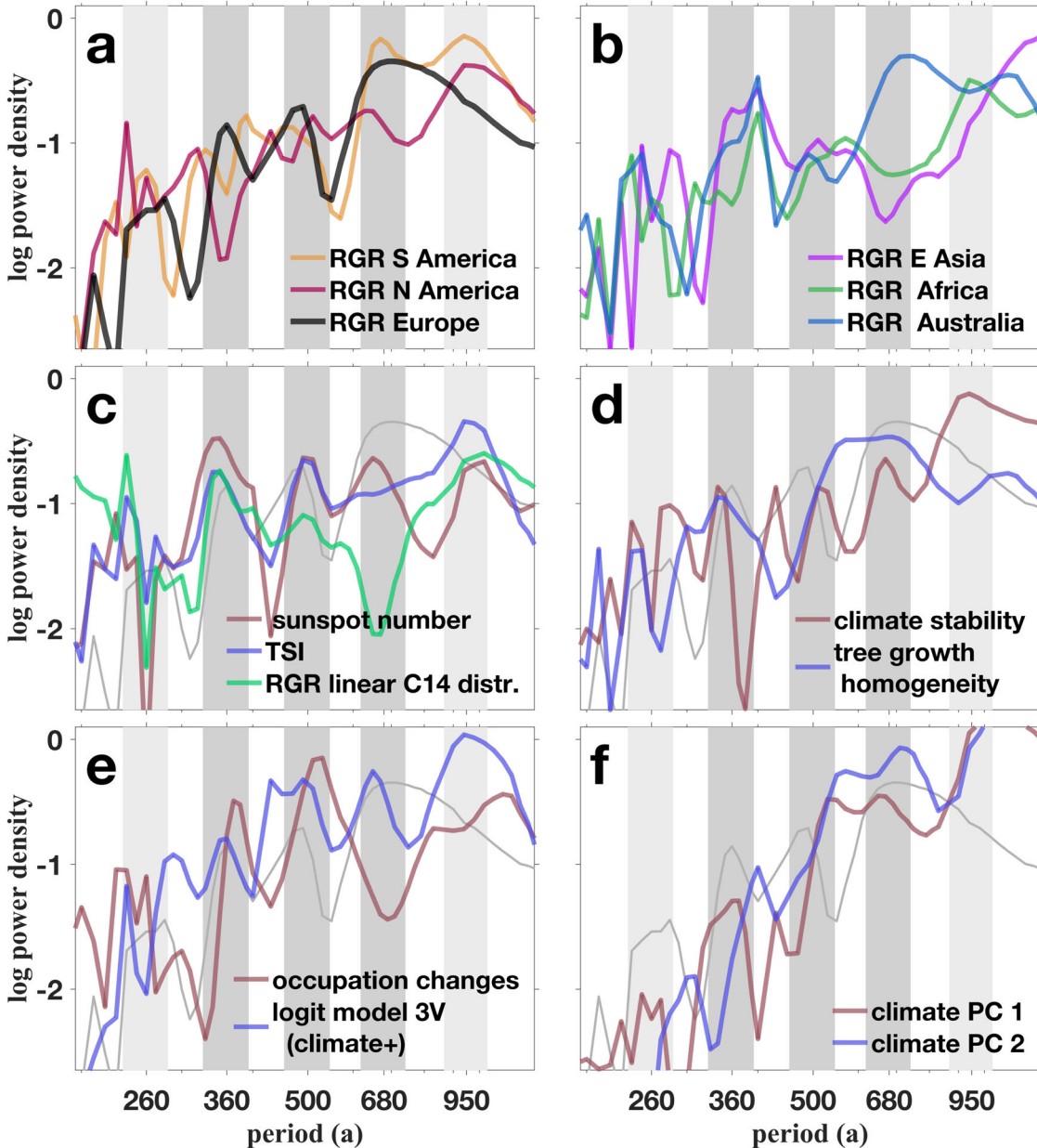

**Fig. 3 | Spectral similarity between growth fluctuations and potential forcings.** Fourier power spectrum of European relative growth rate (RGR, black or grey line) compared to (**a**) RGR of South and North America, (**b**) RGR of East Asia, Africa, and Australia, (**c**) proxies for solar activity: sunspot number[77] and total solar irradiance (TSI), and the artificial RGR of a linear $^{14}$C distribution (calibration effect), (**d**) climate stability index for Europe based on Principal Components (PCs) of paleoproxy time-series and homogeneity in Northern Irish tree growth (reversed standard deviation of tree ring width in bogs) (**e**) boom minus bust density based on independent occupation reconstructions and the full logit model ('3V'), (**f**) first and second PC of 98 paleoclimate proxies. Dark grey shading describes dominant periods of European and non-European RGR, light grey shading additional modes apparent in all climate forcings and in non-European RGRs.

phase overlaps of 73 ± 6% (Fig. 2e–j). In addition, all five dominant frequencies found in the RGR cycles are featured by the TSI power spectrum (Fig. 3c). We excluded that the synchrony of RGR and TSI emerged from calibration artifacts due to the variable slope of the INTCAL curve (Sec. S1). Therefore, the matching of similarity measures suggests a strong link between solar activity and human population growth on large scale.

**Exogenous and endogenous factors in Europe**
For Europe, we not only found by far the highest density of $^{14}$C dates but also a rich account of paleoclimatic information and independent settlement time-series, which both can be compared to the patterns evident in the radiocarbon record. We therefore concentrate on

Europe to help validate inferences about the relationship between the environmental and human past during the Holocene. We collected 98 paleoclimate proxies at decadal-to-centennial resolution for our study period (Tab. S2 and Fig. S1). After statistically adjusting chronologies to improve their synchrony, the time-series were aggregated into principal components (PCs, see Methods). The magnitude of absolute temporal change in those PCs is reversed in sign to obtain a proxy measure for climate stability, which is high at stagnant conditions, including at extremes (Fig. S2). This continental scale and PC-based indicator was used together with the global proxy TSI. In addition to the two climate-related indicators, we considered endogenous processes. For doing so, we inspected the potential of exogenous and endogenous factors using a logistic regression ("logit") model for

boom and bust probability, which includes up to three input variables explored through discrete variants ('1V'–'3V', Table 1). The first input configuration ('1V') entails the same RGR signal shifted ahead in time, which describes an autoregressive contribution. This endogenous and history-dependent contribution to population dynamics is equivalent to the rule that a bust follows a boom after a finite lag – and so does a boom after a bust. In the second configuration ('2V'), the logit model integrates as two exogenous measures solar activity (TSI) and climate stability. Finally, the exogenous and the endogenous forcing were combined to a configuration with three input variables ('3V', Fig. 4, Table 1).

## Climate stability as driver

Our analysis revealed a strong temporal correlation between European population growth and climate stability (80% phase overlap and $r_T = 0.79$, Fig. 5a). The frequency spectra of both solar activity and the continental PC-based indicator for climate stability peak at or close to the three characteristic multicentennial periods found for RGR in Europe and elsewhere (360, 500, and 680a; Fig. 3c). All characteristic periods of continental RGRs from 200 to 950a are inherent to the output of the '3V' logit regression model (informed by climate stability, TSI, and preceding RGR, Fig. 3d–e). In addition to spectral similarity, the '3V' model hindcast evolves highly synchronous with the reconstructed RGR as it fits the timings and durations of all 11+11 European growth busts and booms, which is reflected by 87% phase overlap and $r_T = 0.94$ (Fig. 5). The two skill measures of the combined '3V' variant exceed the ones not only for climate stability alone, but also of the climatic '2V' model variant (82% and $r_T = 0.94$), of the endogenous '1V' one (77% and $r_T = 0.78$, Fig. S3b, c), and of solar variability alone (72% and $r_T = 0.55$, Fig. S3a, skill compilation in Fig. S4).

While the simple ('1V') endogenous lag model already fits the reconstructed growth trajectory very well, several features, such as

busts at 6.4 or 3.5 ka BP, can be better captured by considering climate stability ('2V' and '3V' model variants). The high trim correlations $r_T > 0.75$ and overlaps >75% between RGR on the one side and either PC-based climate stability or lagged RGR on the other side indicate a probable linkage to each explanatory variable, with continental PC-based climate stability being the most probable single factor underlying European growth fluctuations (Fig. S4).

We found a small trim correlation of $r_T = 0.36$ and a phase overlap of 62% between the continental climate stability index and solar forcing over the entire period, including the early Holocene (Fig. S5). However, a causal dependence of stability on multicentennial fluctuations in TSI seems possible after 8 ka BP where overlap and correlation increase to 72% and $r_T = 0.70$, respectively. Similarly high –while not perfect– correlations probably exist for the other world regions, which could explain the degree of phase matching among individual continental RGR cycles or between RGR and global solar forcing.

## Synchronization on sub-continental scales

The dense and relatively uniform spatial distribution of [14]C-dates in Europe (Fig. 1) enables a down-scaling of RGR estimation to regional scales ($10^5$–$10^6$ km$^2$, see Methods). This spatially resolved reconstruction in Fig. S6 in general confirms a mutual independence of regional RGR found in earlier studies[3], as indicated by the generally high heterogeneity. Yet, regional RGR homogenized over nearly the entire European continent twice (at 7.3 and 4.85 ka BP), and recurrently at a sub-continental scale ( >3 · $10^6$ km$^2$). For example, SPDs increased in most of Western Europe but declined in the East in 8–7.9 ka BP, while in 5.6 ka BP, densities increased in the North, and decreased in the Southeast. This sub-continental synchronization of either positive or negative RGR in varying clusters is quantitatively described by the spatial autocorrelation (Fig. S7). Averaged over the study period, spatial autocorrelation of RGR decreases with increasing distance, which agrees with the finding of Freeman et al. for a smaller number of European regions and [14]C-dates[6]. However, this pattern is reversed in some periods of (sub-)continental synchronization when spatial autocorrelation increases with increasing distance, which suggests a recurrent dominance of supra-regional drivers over local processes. Prevalence of exogenous/climatic over endogenous processes at large scales also emerges from computing the synchrony of RGR with solar forcing as a function of spatial extension (Fig. S8). If RGR is averaged for a small area (here Northern Ireland, see below), synchrony is absent (50% overlap is indicative for uncorrelated time-series, see Fig. S4), while phase overlap increases at sub-continental size reaching a

## Table 1 | Input variables and variants of the Logistic Regression Model

| 3V | 1V | time-shifted RGR signal with lag of 350a before 7kaBP and lag of 210a thereafter |
| --- | --- | --- |
| | 2V | climate stability index for Europe based on PCA of 98 paleoclimate proxy time-series |
| | | total solar irradiance (TSI) from[75] |

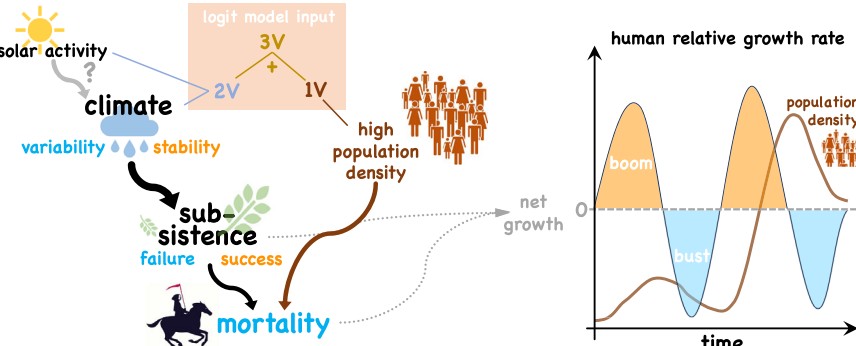

**Fig. 4 | Left:** Conceptual model of human paleodemography. Subsistence success/failure and (partially dependent) mortality determine the relative growth rate (RGR) of human populations. Subsistence, in turn, is influenced by climate stability and the latter probably also by solar activity. During a boom, climate stability favors subsistence success enabling positive and high RGR (orange states). During a bust (blue colors), variations in climatic conditions induce subsistence failures, which in turn, enhance mortality through, e.g., malnutrition. Mortality also increases under elevated population density as a delayed consequence of a boom. Our logit model uses three configurations of input variables: '1V' describes an endogenous control by past population growth and resulting low or high population density. The '2V' variant reflects exogenous control incorporating solar activity and climate stability. The combination of the exogenous the endogenous configurations makes the '3V' variant. **Right:** Idealized growth cycles. Human RGR (black line) alternating between boom (orange shading) and bust phases (light blue) with corresponding population density (brown line).

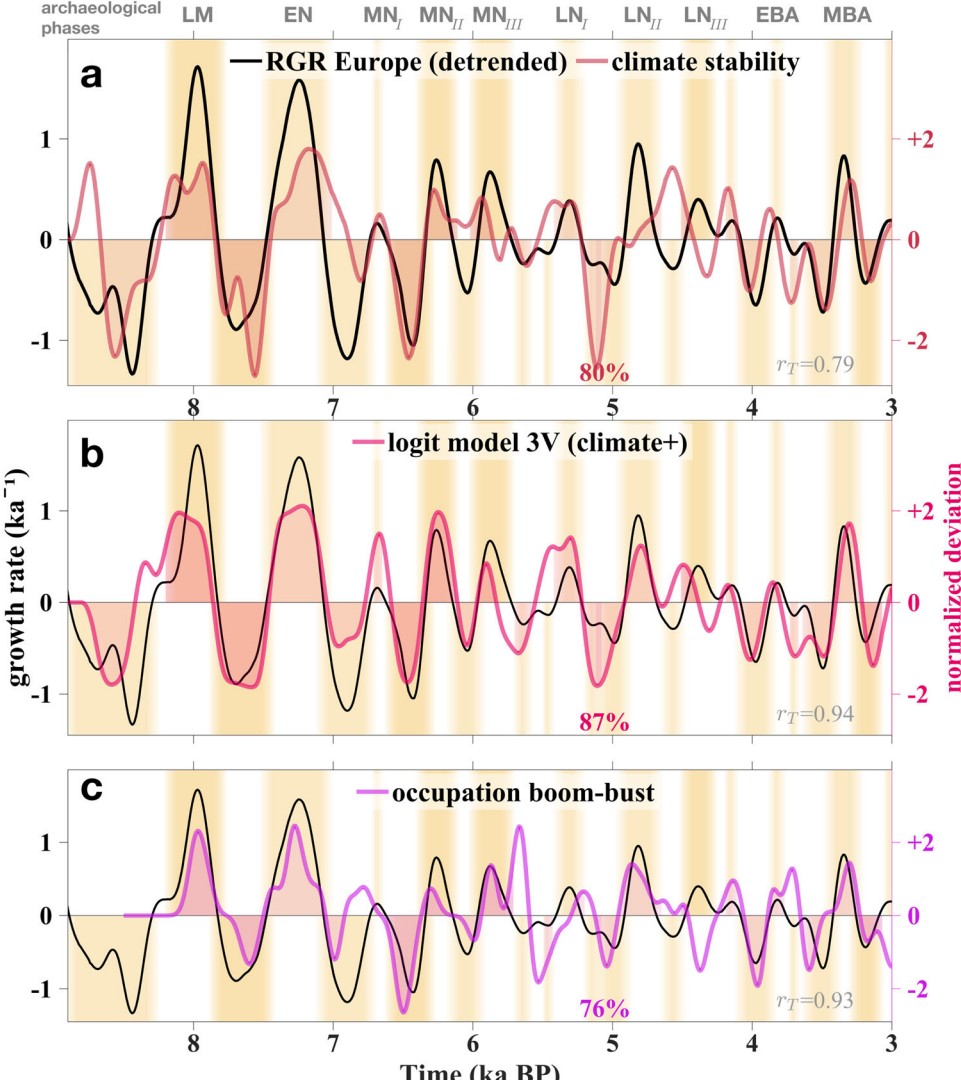

**Fig. 5 | Relative growth rate (RGR, change in SPD) in Europe compared with possible drivers, and independent evidence.** The compared time-series were detrended and normalized (division by standard deviation $\sigma$). Boom and bust periods are shaded to ease the visual check of relative phase overlaps with the respective time-series (colored numbers, see Methods). Grey numbers indicate the trim correlation $r_T$. Boom periods are labelled by archaeological phases as listed in Tab. S3. **a** Climate stability (reversed absolute change in PCs). **b** Probability of a boom minus the one of a bust derived from a logit model based on the three variables (3V) climate stability, total solar irradiance (TSI, Fig. S3a) from[75], and time-lagged growth rate (Fig. S3b). **c** Normalized number of (growth) booms minus the bust number derived from independent archaeological population proxies, mostly occupation density.

maximum of 84% for Western to Central Europe and declines to 72% for entire Europe.

The variable pattern in spatial autocorrelation leads to the quasi-periodic temporal structure of population growth averaged over the continent (Figs. S9, 5, S10). The match between averaged RGR trajectory of the regional reconstruction and the "pooled" continental RGR in Fig. S10 indicates a high robustness of the SPD procedure. The largest exception of the match is visible at the 4 ka BP bust in the down-scaled reconstruction, which mostly relies on a short-term excursion in a single region and thus may be regarded as rather uncertain.

### Confirmation by reported changes in population size
SPD-derived rates of regional population are mostly below 0.5% a⁻¹ (Figs. S6, 6, and S10), well within the rates estimated from independent analyses based on Mid-Holocene house and village structures[21–23]. A population increase over the Holocene[23] is in line with the general increase in probability density 8–4 ka BP (Fig. S9). Previous coarse-resolution reconstructions report a strong positive growth phase in

8.2–7.2 ka BP[23,24], while for this period, our high-resolution European RGR shows two prominent booms and an intermediate stagnation period of vanishing growth, which after trend removal appears as a bust. A decline reported for Southeastern Europe around 6.2 ka BP as well as growth booms around 5.3 ka BP[23,24], or continental booms around 5.9 and 3.8 ka BP[24,25] agree with either continental or respective regional RGR trajectories in Fig. 2d and Fig. S6. Accelerations of growth during 4.5–3.5 ka BP in Southeast and during 4.0–3.5 ka BP in Central Europe[24] correspond to centennial scale regional booms displayed in Fig. S6.

### Multi-proxy approach for reducing uncertainties
To evaluate the precision of our approach, we compiled 20 largely independent (non-SPD) reconstructions of occupation density (see Methods and green boxes in Fig. 1). The calculated probabilities for boom and bust occurrence based on change rates in the independent settlement data agrees with the SPD-based RGR in terms of the three characteristic spectral peaks in Fig. 3e and of a very strong correlation

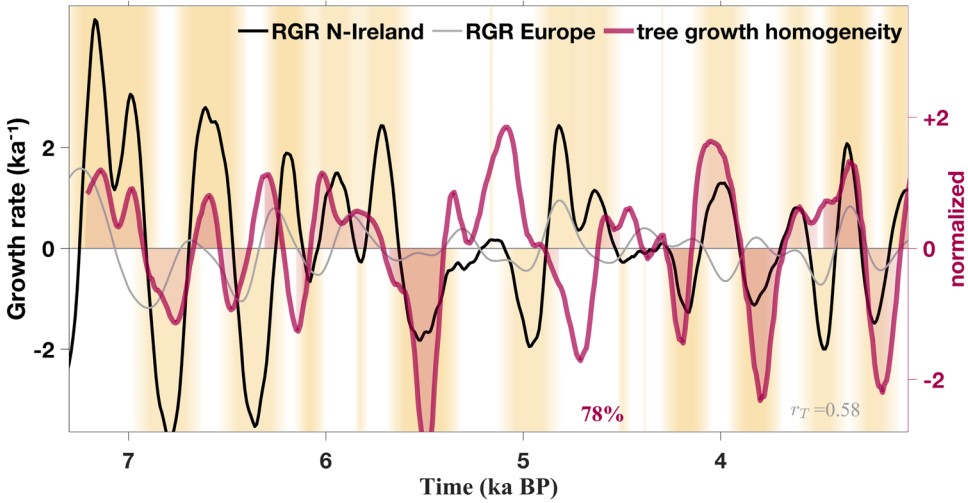

**Fig. 6 | Population growth and environmental stability in Ireland.** Northern Ireland relative growth rate (RGR) compared to the negative anomaly of (spatial) standard deviation of tree ring width ('homogeneity') in Irish bogs as a local proxy for environmental homogeneity and stability.

($r_T$ = 0.93) and phase relationship (76% overlap, Fig. S11a). Peak timings in the two time-series deviate by less than 75 a (Fig. 5b), from which we conclude a high chronological accuracy of our approach.

In addition, all booms inherent to the European growth cycle in Fig. 5 correspond to known archaeological phases listed in Tab. S3. Sub-continental regions with similar growth phases in Fig. S6 include, e.g., Western Europe spatially matching the Atlantic Bronze Age system at 3.3 ka BP, population decline in the Nordic Bronze age from 3.3 to 3.1 ka BP, and population growth at 3.1 ka BP in the Urnfield systems located in Central and Eastern Europe[26].

Reconstruction of SPD and RGR is less uncertain than of climate. Paleoclimatic proxy time-series are well known for inherent problems of precise dating and interpretation (see Methods). Even after integrating an unprecedented number of records for Holocene Europe and developing a Dynamic Time Warping approach for improving chronologies (see Methods and Fig. S1), dating uncertainties remain, also because of lacking reference paleoclimatic time-series. At least, a relatively high degree of explained variance (67%, Fig. S2) by the first five PCs indicates some degree of coherence and reliability of our results.

### Comparison for Northern Ireland at high resolution

We furthermore tested the link between RGR and climate stability for Northern Ireland, where also high-resolution environmental information was available (see yellow box in Fig. 1). Climate stability can be thought to be inherent to the reversed (spatial) variance of annual tree ring widths, which stems from a northern Irish bog record[27]. This tree ring variance, indeed, features multicentennial cyclicity (Fig. S12, Fig. S3d). While the subregional stability/homogeneity index lacks correlation and in-phase behavior with fluctuations in continental RGR ($r_T$ = 0.0, Fig. S12), it agrees well with the RGR of Northern Ireland (78% overlap and $r_T$ = 0.6, Fig. 6), supporting our hypothesis of a linkage between environmental stability and growth also at more local scales.

Against our expectations, but echoing discussion for recent millennia[28], RGR of Northern Ireland often fluctuated in antiphase to tree ring width itself ($r_T$ = − 0.3, Fig. S13), which suggests that conditions beneficial or detrimental to tree growth could have had the opposite effect on human population growth.

### Not climate state, but climate stability matters

A weak to moderate correlation ($r_T^2 \simeq 0.2$-$0.3$) was found at the continental scale when comparing European RGR with the leading PCs for paleoenvironmental conditions (Fig. S14). The power spectra of these indices are dominated by millennial scale variations but also feature contributions at or close to the three characteristic periods of 360 a, 500 a and 680 a (Fig. 3f). Therefore, a sporadic control of climatic states such as found earlier[15] cannot be excluded. However, other drivers including different aspects of climate appear to be more influential on human net growth. All five spectral modes as well as the overall spectral slope of both the continental RGR signals as well as climate stability match the spectral pattern of solar forcing (described by TSI and sunspot number, Fig. 3c). Similarly, phase relationships are high between RGR, stability, and TSI, while moderate for the leading PCs of the climate state. Therefore, (i) a solar trigger of dynamic properties of climate appears possible, and (ii) a dependence of Holocene population dynamics on dynamic properties is more probable than either an exclusive endogenous control or a direct control by climatic states. Almost all paleoclimate−society studies refer to unidirectional climatic shifts, such as towards colder or dryer conditions[14]. Our analysis indicates that any climatic shift regardless of the direction could induce stress to human subsistence and growth. Dry or cold conditions per se may have only moderately influenced subsistence success. This finding resonates with hypotheses on the resilience of societies against climate extremes, but their more subtle susceptibility for long-term climate change[29–32].

Our statistically obvious result implies that human adaptability to changing environmental conditions inferred costs such as societal re-organizations or experimental development of subsistence technologies. Adaptations and their costs depend on predictability, which is constrained by the time scale of variability[33]. Past conditions and fitting societal response repertoires are more difficult to memorize in the long run: changes repeating within a single generation are easier to handle than shifts that re−occurred after > 10−20 generations. Multicentennial variability in solar forcing and Holocene climate (in-)stability probably exceeded a manageable time-scale, then contributing or leading to population recessions. By contrast, higher frequency changes in environmental stressors or singular events may have often stayed within adaptation capacities. Both the autocorrelation analysis and the correlation with climatic changes suggest that exogenous factors could make a small but decisive difference on large spatio-temporal scales.

The most prominent climate anomaly during the Holocene, the global 8.2 ka BP cold event, lags the first massive European bust by >150 a, in agreement with reports of absent or only regional effects on Mesolithic societies in Europe or Southwest Asia during that event[34–36]. The second most prominent event, the 4.2 ka dry event, is

contemporary to positive RGR, especially in Southern Europe. Only sporadic direct imprint of climate on population dynamics was also found for prehistoric Italy[13].

### Endogenous limits of growth—and synergy with stability fluctuations

Population cycles have been extensively studied by ecological theory where several endogenous mechanisms were proposed such as host–pathogen or predator–prey interaction[37,38]. Recently, a predator–prey type model describing warfare dynamics reproduced (idealized) oscillations in Neolithic human populations driven by, e.g., alterations in conflict intensity[9]. Relevance of density-dependent processes is also reflected by a unimodal functional relationship between population size and (positive) demographic growth rates, which was found using global $^{14}$C data similar to the set used here[39].

High population densities after a boom could have intensified mortality processes in past societies, which were connected to inherent failures of social functioning and institutions[40–42]. Such mortality processes comprised diseases[7,43,44] and structural changes in social coherence and political complexity, ultimately leading to disintegration and increases in violent conflicts[9,41,45]. These endogenous mortality processes were often interrelated or mutually amplifying. For example, warfare promoted mobility and, thus, the spread of pathogens.

While endogenous mechanisms for oscillatory dynamics require the cycles to persist over time, the continental and regional growth cycles were intermittently dampened, such as in the half millennium 5.5–5 ka BP (Figs. 5 and 6). Given that the greatest match to continental RGR was achieved by a combination of endogenous and exogenous factors ('3V' logit model), we suggest that the two groups of mechanisms often acted in synergy, in accordance with Moran's theorem in ecology describing an interplay of endogenous and exogenous forces where density-dependent local dynamics synchronizes under a spatially correlated density-independent factor[46]. Higher population densities not only favoured interrelated mortality processes but may have also increased the vulnerability to environmental changes[30,47,48]. Despite occasional or local increases of innovations under high population pressure[49,50], buffer mechanisms such as storage systems, alternative (wild) resources[51,52], or relocation strategies[53] were generally more limited.

The complex interplay of exogenous factors and endogenous mechanisms at multiple spatial scales suggested by our analysis may be tested in modeling experiments.

Already >20 years ago, global human population dynamics in the Holocene has been formalized and simulated at regional resolution by including, e.g., technological evolution, adaptation in and diversity of subsistence styles, migration, and climatic fluctuations[54]. These simulations need to be revised and contrasted to the empirical evidence compiled here or elsewhere as a basis for numerical experiments, which could unravel the quantitative importance of the individual processes underlying social and demographic trajectories.

### Probabilistic exogenous triggers

Mortality factors such as disease and warfare can disperse locally, producing large scale wave patterns. Also growth beneficial processes can be diffusive, such as technological or social innovations arriving with populations from the East before the 8th and in the early 5th millennia BP (S3). Spatial spreading together with the synergy of endogenous mechanisms and exogenous triggers can explain the recurrent emergence of sub-continental autocorrelation synchronous to solar activity and climate stability (Fig. 4).

This interpretation refutes strict environmental determinism while supporting the notion of environmental "possibilism"[55]. Region specific fluctuations in stability still keep a certain phase relationship with their global trigger –here assumed to be solar activity – whereas

regional population dynamics likely were partially decoupled due to endogenous dynamics and mortality/migratory/innovation processes in adjacent regions. Both relationships could explain the lower correlation between climate stability and RGR at the regional scale compared to the continental one as found in our study (e.g., Fig. S7, Fig. S8). In continental scale reconstructions smaller scale processes can be averaged out such as the regional population changes due to inter-regional migration[12] so that the effect of "possibilistic" large scale drivers becomes more apparent.

In light of the remaining dating uncertainties, a solar influence on environmental shifts affecting human growth dynamics is best visible in the spectral domain since all characteristic periods of European, American, African, Australian, or Asian RGR can be found in the solar forcing. Also at longer time-scales, changing solar insolation (TSI) has been proposed as a driver of Pleistocene hunter-gatherer population dynamics[56]. To sharpen a causality relation with TSI has been much facilitated by our growth-based definition of booms and busts. Bevan et al.[12] found that troughs in SPDs for the British Isles coincided with reduced TSI until historic times. However, these minima in TSI are located rather in the late "bust" phase and like other minima actually were synchronous with positive RGR at the transition from low to higher population/activity densities, which confirms the strong correlation between growth and negative TSI found here (Fig. 2e–j). Likewise, Freeman et al.[6] suggested a negligible role of solar forcing on paleodemographic variations, but alike Bevan et al., they used (i) SPDs instead of RGRs, thus introducing incorrect phase shifts, see Fig. 4, (ii) a much smaller data-set, and (iii) conducted their analysis at a smaller scale (see above). These reasons may also explain why Bird et al.[57] could not detect a strong correlation between population stability in North America and model-derived climate stability.

### A global pattern

Our results suggest that a considerable range of subsistence styles including hunting-gathering-fishing, agriculture, or pastoralism could have been similarly sensitive to shifting conditions, especially when population densities were elevated due to a rather stable climate in the preceding 150–350 years. The resulting multicentennial demographic growth cycles were particularly persistent at the continental scale—and beyond: Societies across the globe living in an even greater range of environments than in Europe appeared to follow a similar boom-bust sequence in terms of frequency and phase throughout the Holocene. Non-European boom-bust dynamics may have continued in historic epochs[39]. As continental reconstructions outside Europe rely on a smaller number of $^{14}$C-dates and lack validation with independent demographic proxies, they are thus more uncertain than our growth trajectory for Europe and, thus, require future analyses to confirm the coincidences. Such analyses should include a more global reconstruction of climate stability and its dependence on variations in solar activity, in particular whether and how the atmospheric blocking mechanism during phases of low activity impacts other regional climate systems such as the El Niño Southern Oscillation (ENSO)[58] with consequences for the long term stability in mean climate states (Sec. S2). For Northeast China, a series of cultural/population booms and busts has already been linked to the 500 a ENSO cyclicity[59].

Also, further studies covering the more recent past could verify the existence of recurrent synchronization of external and internal factors proposed here in complex state-level and industrial societies.

## Methods
### Reconstructing human growth rates

Radiocarbon dates derived from the global compilation p3k14c[17] and a series of Europe-centered data sets such as neolithicRC, c14bazAAR[60], EUROEVOL[3,61], and RADON[62] as listed in Tab. S1. We eliminated duplicates with identical $^{14}$C age, $^{14}$C standard deviation (SD$^{14}$C), and geographical coordinates. Boundaries of continental regions in the global

analysis (Fig. 1) reflect the non-uniform spatial density of sites[17]. For European dates, we removed entries with SD14C exceeding max(220a-$\sqrt{8 \cdot 10^5 km^2 \cdot n}$, 40a) with $n$ the regional density of dates within $4^o \times 4^o$ longitude-latitude (see below) so that at very dense 14C date availability such as in UK or the Netherlands only higher precision dates with SD14C around 50a were used. The resulting > 15,000 archeological sites cover nearly entire Europe (Fig. 1). We interpret the radiocarbon dates as a signal for human activity, independently from their affiliation to an archaeological context.

The dates were processed to postcalibration SPDs using the R package RCARBON[63] following the procedure recommended by Crema and Bevan ("calibrate", "binPrep", and "spd"), which referred to the IntCal20 calibration curve[64] (code available at[65]). Calibrated probability distributions were not normalized since normalization can produce artificial spikes in SPD depending on the steepness of the calibration curve[66]. As additional check for the soundness of our methodological choice, we compared the correlation of both normalized and non-normalized SPD with independent occupation density and found a substantial decrease in this correlation with normalization during calibration (Fig. S11b). Data were then binned into 100 a time slices before generating postcalibration SPDs. Sensitivity tests revealed a negligible effect of different bin sizes in the range from 50 a to 150 a.

Causal analysis is facilitated by transforming SPDs to relative growth rates with RGR = {SPD($t$+$\Delta t$)−SPD($t$)}/{SPD($t$)$\Delta t$}. RGRs were obtained without spatial resolution in a simple "pooled" scheme and−in case of Europe−also with regional resolution.

Our spatial approach (for Europe) required outlays of regional units. While as a standard practise in SPD generation, regions are defined referring to specific archaeological/cultural units, we here developed an a posteriori method to create a time-variable region distribution using statistical similarity. Within the wider temporal and spatial scope of our study, borders of cultural entities were not constant nor were cultures persistent in time. Based on SPD processing on a $4^o \times 4^o$ grid, we performed spatial permutation tests of sample sites to detect local deviations in RGR[63,67]. Tests for positive and negative deviations from the RGR distribution under the hypothesis of spatial independence yielded significance levels ($p$ values) and false discovery rates ($q$ values). The $p$ and $q$ values, if below a critical threshold (set to .05), formed a combined score. These scores times the sign of the RGR deviation and the coordinate of locations were fed into the k-means clustering function of R, which requires the number of clusters as an argument. We retrieved an optimal cluster number by minimizing the total within-cluster sum of squares ("withinss") times the weighing factor $1 + \exp\{-10(N_{min}/N^* - 1)\}$, where $N_{min}$ is the size of the smallest resulting cluster and $N^*$ a scaling parameter. By setting $N^* = 120$, we assured that almost all regions contained at least 1000 dates, thus double of the 500 dates considered critical for robust SPD construction[68]. Also, our parameter settings (including critical $p$ and $q$ values) robustly produced spatial clusters in rough agreement with known archaeological entities. The procedure was repeated for each 400 a time slice starting from 3.0–3.4 until 9.4–9.8 ka BP. The geographical outlay of regions was projected onto a $0.1^o \times 0.1^o$ grid, from which we built a spatial continental average. Note that a spatial, area-weighed average differs from an averaging based on relative population sizes since the RGR of a low-density region contributes as much as of an equal size high-density region.

Possible calibration artifacts in all reconstructions were examined as explained in Sec. S1. For evaluating the accuracy of our SPD-based reconstruction, we integrated 20 time-series of alternative proxies for population density covering different time slices from the archeological literature (see Tab. S4). These proxies describe the number of, e.g., settlements or sites in subregions of typically one or two hundred km lateral extension in Europe and western Anatolia (Fig. 1). When

smoothed and normalized change rates exceeded ± 0.5, a boom or bust was attributed, respectively (Fig. S15). Summation over all event series resulted in the density distribution of positive and negative shifts, of which the difference was compared to the SPD-based RGR of Europe.

## Reconstructing climate stability

Our collection of paleoclimatic data started from a previous compilation[18,69] to which we added more recently published records either gathered from the geoscientific data portals www.ncdc.noaa.gov/paleo-search and www.pangaea.de containing also a database of Holocene paleotemperature[70], or digitized from original publications (Tab. S2). The majority of the 98 records located within or close to Europe (Fig. 1) comprised speleothem isotopic oxygen, pollen-based reconstructed temperature, or hydrological indices. The typical dating uncertainty inherent to the time-series was estimated based on studies focusing on the boundary points between the early to mid, and the mid to late Holocene, respectively. A range of about 200 a has been reported for both the start and termination of the 8.2 ka BP event[71], and nearly 500 a uncertainty for the peak of the 4.2 ka BP event[72]. In order to address poorly constrained chronologies and to increase the synchrony between potentially related paleoclimate proxies we applied a dynamic time warping (DTW) algorithm, which adjusted the chronology of each date in each time-series within a conservative range of 150 a. The DTW algorithm provides as a similarity measure between two time-series the shortest path when mapping from one time-series (with index $i = 1, ..., n, n = 98$) onto the other $j$ and has, among others, already been used to increase the alignment of few (two) paleoclimate records[73,74]. Here, we refined the DTW methodology to align multiple records without a predefined target record. First, time-series were split into two parts (>5.8 kaBP, <6.2 kaBP) acknowledging a prominent shift in climate variability during mid-Holocene[18] and increasing the overlap of (partial) time-series as some records started after 9 kaBP or ended before 3 kaBP. Each time-series $i$ is characterized by individual timings with index $k$, thus $T_{i,k}$. For each of the $n(n − 1)/2$ pairs of time-series parts within the respective time window, the smallest (non-normalized) distance in the discretized time space was computed ($\propto \sum |T_{i,k} − T'_{ij,k}|$ with the timings $T_{i,k}$ and $T'_{ij,k}$ of the original series $i$ and warped series $j$, respectively. The DTW distance was adjusted by the relative time-series overlap and amplified by the factor $1 + (d/100)^2$ with $d$ being the maximal distortion in time so that the measure $D_{ij}$ penalizes short overlap and large distortion. $D_{ij}$ then determines the relative weight of each pair $w_{ij} = e^{-(D_{ij}/\overline{D})^2} \cdot w_j$ with $\overline{D}$ denoting the overall mean of $D$ and $w_j$ the DTW weight of sequence $j$ computed at the previous iteration, which starts with $w_j=1/n$ and is set to $w_j = \sum_i w_{ij}/\sum_{ij} w_{ij}$ afterwards. The individual timings $T_{i,k}$ of each time-series $i$ were subsequently displaced by $\sum_j w_{ij} \cdot (T'_{ij,k} − T_{i,k})$, while cutting off absolute displacements of more than 150 a. This way, the timing of events shifted towards the timing of close events in similar time-series. The procedure was repeated 4-8 times until it converged to a set of warped climate proxies with time sequence $T'_{i,k}$. The resulting time-series, plotted together with the original data in Fig. S1, were weighted by $w_j$ to favor higher synchrony among time-series, and fed into a Principal Component Analysis (PCA). Principal Components (PCs) and their respective loads were computed using the function "pca_model" from the MATLAB PCA Toolbox v1.5. From the smoothed averages (see below) of the first five PCs the absolute values of the temporal derivative were summed to express the temporal variability in the climate state. After normalization (subtraction of mean and division by the standard deviation), multiplication by -1 yields an index for slow changes, thus climate stability (Fig. S2). This climate stability index was merged from the two time segments including an interpolation from 5.8 to 6.2 ka BP. Explained variance of the first 3 PCs increased from 39% without DTW to 51% with DTW, and from 54% to 67% for the first five PCs.

A proxy for solar activity derived from a multi-isotope (14C and 10Be) composite for total solar irradiance (TSI)[75]. This proxy well agrees

with other solar time-series[76,77]. The TSI signal was reversed in sign for better comparison with climate stability since a low TSI correlated with high stability (Fig. 2, Sec. S2).

As local but high-resolution proxy of climate stability and homogeneity we integrated tree-ring widths after 7.2ka BP from oaks found in Irish bogs[27,78]. These data record growth for multiple trees deposited up to 70km distance to Belfast[78,79]. The reported spatial variance was normalized and flipped in sign to obtain a description of environmental homogeneity.

## Comparative statistics

All time-series were bandpass-filtered using a distance-decaying filter with 130 a length as high pass filter ('smoothing') and a 1500-year moving-average as low pass filter ('detrending'). Up to three smoothed, detrended, and normalized input variables were fed into variants '1V', '2V', and '3V' of the logistic regression ("logit") model for boom and bust probability as listed in Table 1.

Accounting for the dating uncertainties in paleo-climatic and -demographic time-series, we measured the statistical relation to the target time-series by three different means: (1) a simple Fourier analysis decomposed the time-series into their constituent frequency components. Dominant frequencies (or periods as their inverse values) are visible as peaks in the power spectrum, which can be compared among time-series. For detecting synchrony (matching phase relationship and relative number of common events) we used (2) the trimmed correlation and (3) the phase overlap.

The trimmed correlation $r_T$ is also known as cross-extremogram with zero lag[80]. It measures the similarity of the extremal part of two time-series as only the upper and lower 25% of the distribution were retained before calculation of a Pearson correlation so that low amplitude noise is removed. The exact choice of the percentiles is subjective but was tested to leave no critical impact on the results.

For calculating the phase overlap, time intervals of (positive and negative) peaks in the (RGR) time-series exceeding a small value (thus, booms and bust periods) were extended to the preceding and subsequent 100 a with a weight linearly decreasing from one to zero. The weights were multiplied with the values of the compared time-series having the same sign as the peak and then summed and divided by the total sum. A re-adjustment by a factor close to one made sure that the phase overlap of two identical signals reached 100%. $r_T$ and overlap in general provide similar results (Fig. S4), and their correlation motivated the classification of "strong synchrony" above 67% overlap ($r_T > 0.5$) and "antiphase behaviour" ($r_T < 0$) below 49%.

## Data availability

Links to the individual datasets are provided in Tabs. S1 and S2. Source data are provided as a Source Data in CSV format. The MATLAB and R scripts used to generate the main results together with the collected radiocarbon dates and paleoclimate time-series are available at https://github.com/kaiwirtz3/holocene and in addition archived at https://doi.org/10.5281/zenodo.13748526. Source data are provided with this paper.

## Code availability

The MATLAB and R scripts used to generate the main results together with the collected radiocarbon dates and paleoclimate time-series are available at https://github.com/kaiwirtz3/holocene and in addition archived at https://doi.org/10.5281/zenodo.13748526.

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

## Acknowledgements

We thank Peter Köhler and Jürgen Scheffran for comments on the draft, Ovidio Garcia for suggestions on the methodology, and Mike Baillie for his help in sourcing the Northern Irish bog oak data. **Funding:** K.W., C.L, G.L., and E.Z. were supported by the Helmholtz society via the program "Changing Earth". R.M. was supported by the Irish Research Council through grant 21/PATH-A/9640. N.A. was financed through ANR-DFG grant HA 5407/4-1 ("INTERACT"). J.L. was supported via the Collaborative Research Centre 1266 of the German Research Foundation (DFG, project number 2901391021).

## Author contributions

Conceptualization: K.W. and D.G.; Data curation: K.W., N.A., A.D., J.L., C.L., R.M., D.G.; Methodology: K.W., E.Z., N.A., C.L., R.M.; Analysis & Visualization: K.W.; Writing—original draft: K.W.; Writing—review and editing: K.W., N.A., A.D., J.L., C.L., G.L., R.M., E.Z., D.G.

## Funding

## Competing interests

The authors declare no competing interests.
