## [Transparent Peer Review file · Nature Communications]

Multicentennial cycles in continental demography synchronous with solar activity and climate stability

Corresponding Author: Professor Kai Wirtz

Version 1:

Reviewer comments:

Reviewer #1

(Remarks to the Author)

A major shortcoming of this partially interesting study is the little corroborated correlation between climate stability and population growth rates. What is the hypothesis and rationale behind the oversimplified assumption that varying activity of the sun, if the latter is a main driver of regional climate variability (as stated in line 64 despite of little consensus in the climate community)?

In addition, the reconstruction of the relative demographic trends is not methodologically convincing in all respects and appears to contradict other results in some cases.

Major concerns 1. Nor the climate discussion nor the climate data base appears state of the art (even for an archaeologist lacking broad climate expertise). This is for instance reflected in the referenced papers where the vast majority is older than 2010. Accordingly, recent studies and break throughs are almost completely missed not only in the discussion, but also in the database. Here recent global compilations are available (e.g. Kaufman et al. 2020) which provide a new baseline for this kind of studies, and which integrate most of the data collection here presented together with a much too long and somehow old fashioned reference list of references.

Major concerns 2. Demographic estimates can only partly be estimated using SPDs of large numbers of archaeological radiocarbon date. There are two aspects that make the SPDs appear problematic as a proxy for demographic activities:

a In segments of 'steep slopes', but also 'plateaus' of the calibration curve, the total calibrated values are influenced by the structure of the calibration curve. In this respect, only the unaffected time periods can be considered representative due to these methodological problems. This is also not overcome by producing RGRs because the necessary selection procedure was not carried out in advance.

b ¹⁴C data from archaeological features date deposited remains of human activity. The deposition frequency of prehistoric societies depends on e.g. economic and ritual practices. If these change, the quantity of deposition also changes. Accordingly, SPDs or RGRs can only be used as indications of relative population trends if the economies of societies or ritual practices do not change significantly. This was not taken into account by the procedures which were applied in the analyses.

Major concerns 3. As the SPDs and RGRs still reflect (at least partially) the structure of the calibration curve (cp. MC 2a) and the calibration curve in itself mirrors solar activities, the correlations between the solar and the RGR curves might be labelled a circular argument.

The authors are obviously aware of the problem mentioned in MC2b. Accordingly, they argue that the 'RGR reconstructions would be "well within the rates estimated from independent analyses based on Mid-Holocene house and village structures". But only two studies are cited in favour of this.

Furthermore, 20 examples are cited in Table 4, but 15 of which are also based on the SPD proxy and therefore cannot be used for verification here. In this respect, the correlations found (lines 130-144) are not surprising. In contrast, Results that bring together over 150 studies across Europe that are not based on SPDs and place the archaeological contexts at the centre of the analyses come to completely different conclusions (Müller/Diachenko 2019, Müller 2014, Müller 2015; Schmidt

et al 2021).

Population changes are seen as being caused by technical and social innovations (Schmidt et al. 2021).

Other working groups (cp. Shennan,UCL) use palynological sequences to test the relevance of their SPDs...

A 'method-independent' verification of the RGR reconstructions is therefore not provided in the article. Existing studies that used archaeological arguments to support a population reconstruction tend to contradict the results presented.

Taking these comments into account, the "nature communication" review-questions can be answered as follows:

What are the noteworthy results?

The correlation between "solar activity", climate, subsistence and population density is reduced to a broadening perspective. The reconstructed dependency of population density from solar activity is due to a generalisation of data in a reduction of accuracy.

Will the work be of significance to the field and related fields? How does it compare to the established literature? If the work is not original, please provide relevant references.

Yes. In respect to climate reconstructions the manuscript is not state of the art. In the case of archaeological population reconstructions, those that are not based on SPDs are largely ignored. The compiled data are obviously not compiled by the authors.

Does the work support the conclusions and claims, or is additional evidence needed?

As the reconstruction of population densities is only based on 14C-data, additional evidence is needed. Methodologically independent further evidence has to be integrated and discussed.

Are there any flaws in the data analysis, interpretation and conclusions? Do these prohibit publication or require revision?

Such flaws have been described above.

Is the methodology sound? Does the work meet the expected standards in your field?

The selected climate records have to be added by already existing new compilations. From the supplements it is unclear which 14C-data-sets have been used for which calculation. The reference to general data compilations (e.g. RADON) or references does not provide the reader with the possibility to identify the data which were used. May be, they are in DOI 10.5281/zenodo.10467363 or github.com:kaiwirtz3/holocene (unfortunately not accessible).

Is there enough detail provided in the methods for the work to be reproduced?

See above. As mentioned, unclear, which 14C-data have been used for which analysis (but perhaps in DOI 10.5281/zenodo.10467363).

Tab. 1: RADON lacks <https://radon.ufg.uni-kiel.de/>; RADONb lacks Kneisel et al. 2013

Kneisel et al. 2013: Jutta Kneisel/Martin Hinz/Christoph Rinne, Radon-B. In: <http://radon-b.ufg.uni-kiel.de>

Müller, J. (2015) '8 Million Neolithic Europeans: Social Demography and Social Archaeology on the Scope of Change – from the Near East to Scandinavia' in Kristiansen, K., Smejda, L. and Turek, J., eds., Paradigm Found, Oxford: Oxbow, 200-214.

Müller, J. and Diachenko, A. (2019) 'Tracing long-term demographic changes: The issue of spatial scales', Plos one, 14(1), e0208739.

Schmidt, I., Hilpert, J., Kretschmer, I., Peters, R., Broich, M., Schiesberg, S., Vogels, O., Wendt, K. P., Zimmermann, A. and Maier, A. (2021) 'Approaching prehistoric demography: proxies, scales and scope of the Cologne Protocol in European contexts', Phil. Trans. R. Soc. B 376: 20190714. <http://dx.doi.org/10.1098/rstb.2019.0714>.

(Remarks on code availability)

Reviewer #2

(Remarks to the Author)

Scholars in the "History of Climate and Society" – the transdisciplinary field that identifies human responses to past climate change – have long argued that climatic trends and anomalies played a crucial role in the history of human demographics. This argument follows two tracks. The first focuses on the Pleistocene, identifying the epoch's dramatic swings in climate as a key force behind human mortality and migration. The second concentrates on distinct societies in the Holocene, suggesting that climate change played a key role in the collapse of those societies, and in turn episodes of regional depopulation.

Yet until now, scholars have found no convincing connection between the climatic oscillations of the Holocene and the population of entire continents, let alone the globe. This makes intuitive sense: in most regions the climate of the Holocene was relatively warm, wet, and stable, compared to that of the Pleistocene, and a raft of buffering mechanisms – from

irrigation to granaries – debuted with the development of so-called “complex civilizations.”

Yet by combining reams of diverse archaeological and paleoscientific evidence, this manuscript argues that by focusing on transitions to different climate states, scholars have missed a sweeping connection between human demographic change and shifts in climate stability. The authors claim that periods of climatic stability correlate with cycles of human population growth and regrowth across all six inhabited continents, while periods of instability are associated with population declines. They find that variability in total solar irradiance is to blame for changes in climate stability, which would suggest that human history is profoundly tied to – in fact, to some extent determined by – the shifting environments of the Sun.

If the authors are correct, they could transform how scholars of the past understand the broad sweep of human history. In all likelihood this would be a manuscript of fundamental importance: certainly worthy of publication in *Nature Communications*.

As environmental historians whose work is largely qualitative in nature, and concentrates on the history of the past 1,500 years, we should acknowledge that we are not well equipped to evaluate some of the methods employed in this manuscript. We do not know whether summed probability distribution, for example, is an appropriate method to use when evaluating the 91,000 archaeological radiocarbon dates amassed by the authors, and indeed we are not sure whether the dates are a good way to reconstruct paleodemography.

Yet we can evaluate how this manuscript broadly conceptualizes human and climatic histories, and how it makes causal claims or handles specific evidence that falls squarely in our areas of expertise. Although we are inherently skeptical of studies that identify correlations between environmental and human changes across large spatiotemporal scales – a function in part of our discipline, which is more likely to emphasize contingency than correlation – we find that this manuscript for the most part skillfully handles those topics that we know well.

Beginning on line 170, for example, the authors claim that “Against expectation, RGR of Northern Ireland often fluctuated in antiphase to tree ring width itself... which suggests that conditions beneficial or detrimental to tree growth could have had the opposite effect on human population growth.” Here, the authors should consult B. Campbell and F. Ludlow, “Climate, Disease and Society in Late-Medieval Ireland” *Proceedings of the Royal Irish Academy* 120 (2020). Campbell and Ludlow indeed note that conditions in this region favorable to tree growth are often inverse to those favorable for crops. This helps to explain the authors’ finding, which is not at all “against expectation” to regional experts. The authors should therefore tweak their wording, but it is significant to us that they accurately and independently made this rather obscure finding.

More broadly, we note that some historians have long argued that shifts in climate stability have influenced human history at least as much as changes in climate state. In 1980 Jan de Vries, for example, argued that climate stability mattered more for European farmers than changes in climate state (see Jan de Vries, “Measuring the impact of climate on history: the search for appropriate methodologies.” *The Journal of Interdisciplinary History* 10, no. 4 (1980)); in 2018 Dagomar Degroot made similar claims with reference to the Dutch Republic (Dagomar Degroot, *The frigid golden age: Climate change, the little ice age, and the Dutch Republic, 1560–1720*. Cambridge University Press, 2018). The scale considered by these historians is far smaller in time and space than that analyzed in this manuscript. Yet to us it does seem telling that processes on a relatively small scale seem to confirm the claims made in this manuscript that concern a much larger scale. We would suggest that the authors briefly acknowledge this apparent confirmation.

In general, we were impressed not only by the sheer quantity of evidence amassed by the authors, but also by how the authors critically evaluate their evidence and their methods for analyzing evidence. The authors emphasize potential shortcomings in their sources and methods, and they acknowledge critical scholarship that, for example, identifies problems with the radiocarbon dating of archaeological remains as a proxy for human demographics. The authors do well to evaluate the strengths and limitations of the paleoclimatic reconstructions they employ, though we think that not all of the proxies examined in this manuscript can be characterized as “high resolution” (line 70).

We do think, however, that this manuscript is less accessibly written than many publications in *Nature Comms*. Not only are the manuscript’s descriptions of complex, correlated changes unnecessarily wordy in places, but the manuscript’s argument is not expressed with sufficient clarity in its first two pages. Instead, the argument unwinds slowly as the authors introduce data and acronyms. We suggest that the authors succinctly provide their argument on the first or second pages of their manuscript, and then explain their evidence and methods.

We suggest, therefore, that this manuscript be published in *Nature Comms*, with minor revisions. However, we note again that we are not able to evaluate some of the manuscript’s evidence and methods.

(Remarks on code availability)

Reviewer #3

(Remarks to the Author)

Two overarching and substantial issues need to be raised at the onset:

1. in the paper the use of boom/bust scenarios (e.g. line 46 but also the rest of the paper) does not follow convention and therefore introduces significant confusion in the authors’ interpretations. RGRs are estimated as relative growth rates with

$[RGR = \{SPD(t+\Delta t) - SPD(t)\} / \{SPD(t)\Delta t}]$ - a calculation discretely tucked away in the methods section. There is precedent in the use of growth rates to analyze 14C demographic proxies (Arroyo-Kalin & Riris 2021, RTSB). But, growth rates do NOT IDENTIFY booms and busts. They are NOT EQUIVALENT to Boom or Busts either. This is shown by the authors themselves in Figure S.8. In most previously published SCPD studies, a boom is defined as overshoot above the expected demographic growth model; a bust is falling below the floor of the expected demographic growth model. RGRs are neither, as shown in Fig. S8. Each RGR upturn or downturn is an acceleration or de-acceleration of growth rates, nothing else until it is shown that it expresses a time-series that exceeds (+ or -) the expected demographic model.

This does not invalidate the point the authors wish to make but calls for more clarity and distinction between upswings, downturns, and overshooting above and below the demographic model against which SCPD are being assessed. Co-patterning with solar activity is significant. That the latter dictates the condition for population boom and busts is not demonstrated by the paper.

2. The section S1 "Solar influence on climate fluctuations" is too important for the overall reasoning presented in the paper to be buried in SM. A summary of its key points deserves to be part of the introduction of the paper or be included in the section "Synchrony with solar forcing". Moreover, S1 ends with an insufficiently substantiated statement that verges on hasty generalization (406): "Based on this finding, we here hypothesize that lower solar activity may on a short timescale enhance the likelihood of weather extremes but on a long timescale reduce variations in the mean climate state". This hypothesis is never tested in the paper. I can see how rapid downturns in solar activity may have short-term effects on weather extremes (e.g. through atmospheric blocking patterns, as suggested by authors) but their influence on the long-term mean climate state is less clear: a compounded effect over the long-term is likely an outcome of rhythm and frequency of rapid downturns, not a long-term property per se.

Others:

Line 39 - If comparisons of continental datasets appear to be poorly synchronised (paragraph 1), how would focussing on a single region be a legitimate way to generalise to "validate any global inferences ... at continental scales" - another hasty generalization?

Lines 70, 71: both statistical adjusting of chronologies to improve synchrony and the ways in which time-series were aggregated into principal components are difficult to follow in SM.

Lines 78-83. Models IV, 2V, 3V - all unclearly explained in the paper. Particular, Fig. 3 (left) is difficult to make sense of. In S1 it is suggested that lower solar forcing would impact short term climate by inducing more extreme conditions (this made sense to me). Fig 3, Left would seem suggests the opposite (high solar activity-> variability-> failure, light blue arrows), and conversely? Can this be improved for the sake of communicating more effectively. In addition, pathway 2V should be described in words (high solar activity leads to climate variability which leads to climate failure) and it should be reconciled with discussion of short-term versus long-term effects (S1, line 406) reconciled

Lines 88-90: This highest spectral agreement with the '3V' logit regression model is hard to understand.

Line 97: These three population "collapses" cannot be seen as such on the basis of RGRs. There's no collapse, just de-acceleration. Shennan et al. 2013 show clearly that a "proper" bust in one European region is not reflected in another (see their Figure 3, especially data for French and German regions for the ~6.4 BP chron). The differential impact of arrival of Neolithic agrarian societies across the European region is, furthermore, not considered (Bevan et al. 2017, PNAS)

Figure S7 hardly supports a population bust at c. 4000 BP

Line 104: Can the authors check the underlying for Fig. S6 ? Time shifting the 'Climate stability' curve by about 1000 years seems to provide a much better match between stability and solar forcing, which would agree much better with the suggestion that short-term lower solar activity directly impacts climate stability.

Line 239: Most of the time, and especially with RGRs as opposed to Boom / Bust scenarios, what is more clearly tracked is de-acceleration in birth rates rather than large scale factors acting rapidly. RGRs do not provide any information about population decimation on their own.

(Remarks on code availability)

Reviewer #4

(Remarks to the Author)

(Remarks on code availability)

Version 2:

Reviewer comments:

Reviewer #2

(Remarks to the Author)

The author(s) have responded to our suggestions, implementing them skillfully or else rejecting them with an explanation that satisfies us. They have also revised their manuscript so it is more accessible than it was.

We recommend citing Degroot 2018 ("The Frigid Golden Age") rather than Degroot et al. 2021 ("Towards a Rigorous Understanding) for the point on climatic variability being more important than trends in some historical work.

Otherwise, we have little to add. We still believe that this manuscript should be published in Nature, but we note again that our expertise does not encompass all of the data and methods used in this study.

(Remarks on code availability)

Reviewer #4

(Remarks to the Author)

(Remarks on code availability)

Reviewer #5

(Remarks to the Author)

I was asked to review this paper after a round of reviewing that seems to have led to a rejection if I read the last line of the rebuttal letter correctly.

What are the noteworthy results?

The authors demonstrate a relationship between climate variability and fluctuations in relative population growth rates at scales from the regional to the continental and suggest that climate variability may be related to fluctuations in solar activity. They develop a model that includes both climate variability and endogenous processes.

Will the work be of significance to the field and related fields? How does it compare to the established literature? If the work is not original, please provide relevant references.

The work is of wide-ranging significance to archaeology and other palaeosciences. It goes well beyond the extensive previous work attempting to account for variations in past human demographic rates and takes it to a higher level.

Does the work support the conclusions and claims, or is additional evidence needed?

Yes, the work convincingly supports the conclusions

Are there any flaws in the data analysis, interpretation and conclusions? Do these prohibit publication or require revision? Is the methodology sound? Does the work meet the expected standards in your field?

These were criticised by previous reviewers. I found the detailed responses to the reviewers presented in the rebuttal letter and the measures they have taken in the paper to address them very convincing. The authors have gone to enormous lengths to address potential issues with their methods and data sources and their effect on their argument. This goes well beyond the usual standards in the field. In my view the data analysis, interpretation and conclusions are well-founded.

Is there enough detail provided in the methods for the work to be reproduced?

Yes

(Remarks on code availability)

Reviewer #6

(Remarks to the Author)

Review of "Multicentennial cycles in continental demography synchronous with solar activity and climate stability"

Summary and recommendation:

Dear Editor,

Thank you for the opportunity to review "Multicentennial cycles in continental demography synchronous with solar activity and climate stability." I do not recommend publication without significant revisions. I note the substantive weaknesses/omissions of the paper below. These weaknesses undermine contribution #2 below. These should be addressed so that the paper can make a significant contribution to the fields of archaeology, human population ecology, and beyond.

The current manuscript is really exciting and reports a conceptual research design that will allow the scientific community to explain the long-term drivers of human population dynamics across scales. The manuscript reports on the synchrony of population increases and decreases with the intensity of solar energy and climate stability. The analysis covers multiple scales, including continental comparisons, a comparison of regions in Europe, and a single Irish case study. The authors find that population growth rates are locked in phase with the intensity of solar energy and climate stability. The authors spend a lot of time checking the redundancy of this finding across scales, with a specific focus on Europe. The authors conclude that there are highly general population dynamics across modes of subsistence production (something found by previous global and continental comparisons) and that researchers need to focus more on climate stability rather than extremes when understanding the ability of populations to cope with climate/ecological change.

Potentially significant contributions of the paper include:

- 1) Data synthesis of archaeological radiocarbon and paleoclimate data. They build on existing data sets to develop the ability to engage in better global comparisons.
- 2) The manuscript reports interesting results that global archaeological radiocarbon time-series vary in phase with solar radiation and climate stability across continents. These results could help us understand the long-term drivers of human population growth and decline.

I wish the authors the best of luck with their exciting research program and on revisions.

Sincerely, Jacob Freeman

p.s. As a rule I generally do not cite my own work in reviews. However, it seems unavoidable in this case given the high degree of relevance. I will note that I have nothing to gain from another couple of citations because I have already been promoted. That also means I am not as political in my comments as I used to be (:

Substantive Comments:

1) This paper is part of a growing literature of comparative human population ecology during the Holocene. Yet, the author does not acknowledge nor, more importantly, really build on this literature to illustrate how this study advances our knowledge of long-term human population dynamics. This has two consequences. (a) It stagnates the critical discussion of ideas. (b) I believe that this contributes to the lack of a coherent theoretical framework within which to ground the results of the current paper.

(a) The current manuscript claims that this study is almost completely new in scope and topic. And this is a disappointing strategic decision by the author because it stagnates the critical discussion of ideas that is central to knowledge growth in science. For example, Freeman et al. 2018 study the synchrony of radiocarbon SPDs with each other and with solar energy at a global scale. The current paper appears to build on the conceptual framework laid out in the Freeman et al 2018 paper. For instance, Freeman et al. state:

"The synchrony of energy consumption among human societies, at a global scale, could result from two global mechanisms. First, human societies may all respond similarly to fluctuations in an external driver—the so-called Moran effect (18) Thus, we might expect that fluxes in solar energy cause human populations to synchronize, and, if so, human populations in different biophysical environments should synchronize with each other and with the influx of solar energy.... Second, direct interactions such as trade and migration, as well as indirect interactions (e.g., common disease vectors or indirect trade), may cause the synchrony of energy consumption among human populations. (p. 9963)."

In sum, either external or internal drivers may cause human populations to synchronize. The current paper clearly follows on this conceptual investigation and advances on the earlier study by analyzing a lot more data. Indeed, the 2018 paper was the impetus for the P3K radiocarbon data set so that studies such as the current manuscript could be conducted at a truly global scale. Interestingly, the current paper and the 2018 paper come to different conclusions, and it is important to understand why they come to different conclusions. It would be really exciting and interesting if solar forcing were driving synchrony between human populations.

I suggest that the author re-frame the paper as an advance on earlier studies and the growing literature engaging in global comparisons of long-term human population dynamics. Honestly discussing these earlier works and how this study advances on them is a more powerful framing than claiming to be the first to study such issues.

A few papers that engage in comparative studies of archaeological radiocarbon.

@article{freeman2024long,
title={The long-term expansion and recession of human populations},

author={Freeman, Jacob and Robinson, Erick and Bird, Darcy and Hard, Robert J and Mauldin, Raymond P and Anderies, John M},
journal={Proceedings of the National Academy of Sciences},
volume={121},
number={12},
pages={e2312207121},
year={2024},
publisher={National Acad Sciences}
}

@article{riris2024frequent,
title={Frequent disturbances enhanced the resilience of past human populations},
author={Riris, Philip and Silva, Fabio and Crema, Enrico and Palmisano, Alessio and Robinson, Erick and Siegel, Peter E and French, Jennifer C and J{o}rgensen, Erlend Kirkeng and Maezumi, Shira Yoshi and Solheim, Steinar and others},
journal={Nature},
pages={1--6},
year={2024},
publisher={Nature Publishing Group UK London}
}

@article{bird2020first,
title={A first empirical analysis of population stability in North America using radiocarbon records},
author={Bird, Darcy and Freeman, Jacob and Robinson, Erick and Maughan, Gideon and Finley, Judson Byrd and Lambert, Patricia M and Kelly, Robert L},
journal={The Holocene},
volume={30},
number={9},
pages={1345--1359},
year={2020},
publisher={SAGE Publications Sage UK: London, England}
}

@article{freeman2018synchronization,
title={Synchronization of energy consumption by human societies throughout the Holocene},
author={Freeman, Jacob and Baggio, Jacopo A and Robinson, Erick and Byers, David A and Gayo, Eugenia and Finley, Judson Byrd and Meyer, Jack A and Kelly, Robert L and Anderies, John M},
journal={Proceedings of the National Academy of Sciences},
volume={115},
number={40},
pages={9962--9967},
year={2018},
publisher={National Acad Sciences}
}

@article{jorgensen2022climatic,
title={Climatic changes cause synchronous population dynamics and adaptive strategies among coastal hunter-gatherers in Holocene northern Europe},
author={J{o}rgensen, Erlend Kirkeng and Pesonen, Petro and Tallavaara, Miikka},
journal={Quaternary Research},
volume={108},
pages={107--122},
year={2022},
publisher={Cambridge University Press}
}

(b) One of the main weaknesses of the paper is that it lacks an overall theoretical structure. This contributes to the paper not reading very well, as noted by previous reviewers. I think that engaging with some of the previous studies of global scale comparisons noted above can help the author develop a better narrative structure that critiques and builds on previous work. For example, in the current version of the manuscript, the author launches into a discussion of the importance of climate stability, but this comes off as a purely inductive exercise in which the author is searching for associations, and then post-hoc justifying why they are meaningful. There is good theory in the resilience literature that would suggest that climate uncertainty is much more difficult for humans to deal with than predictable extremes. This is because humans use rules and infrastructure (culture) to create flows of resources, but inevitably the rules and infrastructure must be designed for a given climate regime. If climate uncertainty increases, this will stress existing infrastructure systems and social rules for allocating resources.

See for example:

27. Carpenter SR, Brock WA, Folke C, van Nes EH, Scheffer M (2015) Allowing variance may enlarge the safe operating space for exploited ecosystems. *Proc Natl Acad Sci USA* 112:14384–14389.

28. Freeman J, Peeples M, Anderies JM (2015) Toward a non-linear theory of the transition from foraging to farming. *J Anthropol Archaeol* 40:109–122.

29. Anderies JM (2006) Robustness, institutions, and large-scale change in social-ecological systems: The Hohokam of the Phoenix basin. *J Institutional Econ*

2(2):133–155.

2) In general, the author has effectively commented on previous reviewer critiques. However, a critical part of the methodology of this paper is not well presented and requires revision for publication. The authors do a good job of describing how they constructed the SPDs and RGR. They also describe how they reconstruct paleoclimate stability fairly well.

The major short coming is that the authors do not describe their methods for assessing synchrony nor identifying the dominant modes. I tried to figure it out from the r and matlab code, but these files are unorganized and not commented very well, thus, I could not figure it out in a reasonable amount of time.

So how do you calculate phase overlap? It seems from Fig. 1 that this is done by converting the time series using some kind of transform (Fourier transform??) as assessing the degree of locking (as on a unit circle). This is really important to make clear because using a method that assumes that the underlying variables are continuous and not noisy will artificially inflate the degree of phase locking. I have no way of assessing the robustness of the current results to the choice of method. I can at least say that the trimmed Pearson coefficient, even in the de trended data, is probably not the best measure of association.

In fact, the original title of the 2018 paper noted above was “Solar energy drives human population synchrony.” However, after a deep review of how to assess the synchrony of noisy and discrete data, the synchrony between the SPDS and solar energy weakened considerably across scales. The SPDs are discrete and noisy data, no matter how much they are smoothed. I can feel a response coming here, so let me just say this:

At minimum, the paper would improve significantly if the author presents their methods for calculating the phase overlap of the time-series. They should discuss why their method is more appropriate than say, mutual information with a Markov process to construct simulated time-series. Even better would be to compare the methods that they use and the methods that we describe in our 2018 paper. If the results are robust to changing method, then great! If not, then the discussion of the appropriate methods becomes really important to have in the literature so that other researchers can assess and build better methods.

See also:

Cazelles B (2004) Symbolic dynamics for identifying similarity between rhythms of ecological time series. *Ecol Lett* 7:755–763.

Minor Comments/Reading notes:

1. Abstract: “A possible link from multicentennial variations in solar activity to climate stability as a critical condition for human subsistence success, could explain the occurrence and partial phase synchrony of growth cycles worldwide.”

What is meant by subsistence success here? I don’t understand this sentence and general. Please clarify.

2. line 24 “However, going back into the past and focusing on local to regional scales, the archaeological record discloses a more disruptive picture with a series of rises and falls of societies connected to concomitant shifts in population density.”

This is a statement of fact not backed up by any evidence. Add citations to works that demonstrate this fact. Perhaps describe an example of the general pattern that you have in mind.

3. “but human demography and climatic drivers seem to lack spatio-temporal coherence at larger spatial scales.”

Stated as fact with no evidence to back the claim. Add citations to works that demonstrate this

4. line 130 “The causality of prehistoric demographic change also remains obscure because very few studies have tackled the problem at the continental scale, and none assessed the climate–society relationship systematically at regional scales.”

This is hard to follow. I am aware of many studies that tackle the problem at a continental scale and assess the climate–society relationship systematically at regional scales. In addition to the Lima et al. 2024 paper cited, here are just a few examples:

@article{freeman2024long,
title={The long-term expansion and recession of human populations},
author={Freeman, Jacob and Robinson, Erick and Bird, Darcy and Hard, Robert J and Mauldin, Raymond P and Anderies, John M},
journal={Proceedings of the National Academy of Sciences},
volume={121},
number={12},

```

pages={e2312207121},
year={2024},
publisher={National Acad Sciences}
}
@article{riris2024frequent,
title={Frequent disturbances enhanced the resilience of past human populations},
author={Riris, Philip and Silva, Fabio and Crema, Enrico and Palmisano, Alessio and Robinson, Erick and Siegel, Peter E and French, Jennifer C and J{\o}rgensen, Erlend Kirkeng and Maezumi, Shira Yoshi and Solheim, Steinar and others},
journal={Nature},
pages={1--6},
year={2024},
publisher={Nature Publishing Group UK London}
}
@article{bird2020first,
title={A first empirical analysis of population stability in North America using radiocarbon records},
author={Bird, Darcy and Freeman, Jacob and Robinson, Erick and Maughan, Gideon and Finley, Judson Byrd and Lambert, Patricia M and Kelly, Robert L},
journal={The Holocene},
volume={30},
number={9},
pages={1345--1359},
year={2020},
publisher={SAGE Publications Sage UK: London, England}
}
@article{freeman2018synchronization,
title={Synchronization of energy consumption by human societies throughout the Holocene},
author={Freeman, Jacob and Baggio, Jacopo A and Robinson, Erick and Byers, David A and Gayo, Eugenia and Finley, Judson Byrd and Meyer, Jack A and Kelly, Robert L and Anderies, John M},
journal={Proceedings of the National Academy of Sciences},
volume={115},
number={40},
pages={9962--9967},
year={2018},
publisher={National Acad Sciences}
}
}
@article{jorgensen2022climatic,
title={Climatic changes cause synchronous population dynamics and adaptive strategies among coastal hunter-gatherers in Holocene northern Europe},
author={J{\o}rgensen, Erlend Kirkeng and Pesonen, Petro and Tallavaara, Miikka},
journal={Quaternary Research},
volume={108},
pages={107--122},
year={2022},
publisher={Cambridge University Press}
}
}

```

5. Line 40 ``Our usage of RGR allows a more precise causal inference analysis than referring to, e.g., extrema in SPDs because drivers act on growth rate rather than on population density. Analogously, 'booms' and 'busts' in archaeology are regularly defined based on peaks and troughs in population size, so that the second half of, e.g., a "boom" necessarily features negative RGR. However, a high but declining population size more likely describes a crisis situation than a low but rising population. Therefore, we here distinguish (growth) booms and busts according to the sign of RGR."

This just seems like an odd place for this text. As a reader, I am still trying to understand why this study is important and what it will report, and then I am thrown into a detailed methodological discussion. I suggest moving this text to the methods section.

6. ``Apart of the period 6.5 to 5.5 ka BP, the demographic trajectories of Europe and South America show a high level of agreement."

Check—should it be ``Apart from???"

7. 160: "If the calibrated probability distribution were normalized, trim correlation between reconstructed SPD-based RGR and changes in occupation density would decrease from nearly one to zero (Fig. S11b, Fig. S5), which approves our methodological choice of not normalizing during calibration."

The logic here is difficult to follow. Move all discussion of normalized SPDs to the SI. It would seem that your choice to normalize or not normalize the SPDs affects the results. This does not “approve” the non-normalization, it is just a fact. The results are not robust to changes in method (normal vs non-normalized). You need to make an independent argument that non-normalized are more appropriate than normalized SPDs (but please make the argument in the methods and SI rather than here).

8. “Our analysis indicates that any climatic shift regardless of the direction could induce stress to human subsistence and growth. Dry or cold conditions per se may have only moderately influenced subsistence success. This finding resonates with previous hypotheses on the resilience of historic societies against climate extremes, but their more subtle susceptibility for long-term climate change [32, 33].”

9. Line 221 “no or” typo.

10. I cannot access the data and code to review via the links. Both links below appear broken. I had to search for kaiwirtz3 on Github and click on the user profile to find the data and code. The code is not really commented at all, thus it is very difficult to follow and replicate the author’s procedures.

<https://github.com:kaiwirtz3/holocene>

DOI: 10.5281/zenodo.10467363

(Remarks on code availability)

The links provided in the paper do not work. I had to search the user name on GitHub. The code is not well organized and minimally commented. This will make replication difficult?

Version 3:

Reviewer comments:

Reviewer #6

(Remarks to the Author)

The authors have addressed my concerns. The paper will, I hope, generate continued research on the topic of human population synchrony in archaeology.

(Remarks on code availability)

It is OK.

Nature editorial offices
The Campus
4 Crinan Street
London N1 9XW
United Kingdom

Institute of Coastal Systems
Analysis and Modeling
Ecosystem Modeling

Head of Department
Prof. Dr.
Kai W. Wirtz
T +49 4152 87-1513
M +49 160 75 77 327
kai.wirtz@hereon.de

Rebuttal NCOMMS-24-01345

June, 6 2024

Thank you for handling our manuscript “Multicentennial cycles in continental demography synchronous with solar-triggered climate stability” (NCLIM-21050827). In your decision dated May 29, 2024, you referred primarily to Reviewer #1's criticism regarding “summed probability distributions to reconstruct demographic trends and how up-to-date climate discussion is.” As we will show below, Reviewer #1's concerns appear to be based on misperceptions rather than actual weaknesses in our approach.

1) “Nor the climate discussion nor the climate data base appears state of the art ... This is for instance reflected in the referenced papers where the vast majority is older than 2010. Accordingly, recent studies and break throughs are almost completely missed not only in the discussion, but also in the database. Here recent global compilations are available (e.g. Kaufman et al. 2020) which provide a new baseline for this kind of studies, and which integrate most of the data collection here presented together with a much too long and somehow old fashioned reference list of references.”

Climate proxy compilations select data based on specific purposes. Kaufman et al. (2020) aimed at reconstructing global Holocene temperature and collected proxies with a known relationship to temperature. We targeted climate variability and selected proxy time-series with higher temporal resolution and coverage than most K2020 entries. While we do not believe that the age of publication in paleoclimatology strongly correlates with data quality, we checked the statistics: the average publication year for the K2020 database for Europe is **1995**, while ours is **2010**. Therefore, we cannot comprehend how “a vast majority [of our records] is older than 2010” or how we are behind “state-of-the-art.” Our compilation is accessible in a popular repository. No rationale is given for why the discussion of solar influences on climate, co-authored by leading experts in the field, isn't up-to-date.

2) “Demographic estimates can only partly be estimated using SPDs of large numbers of archaeological radiocarbon date. There are two aspects that make the SPDs appear problematic as a proxy for demographic activities:

a In segments of ‘steep slopes’, but also ‘plateaus’ of the calibration curve, the total calibrated values are influenced by the structure of the calibration curve. In this respect,

only the unaffected time periods can be considered representative due to these methodological problems. This is also not overcome by producing RGRs because the necessary selection procedure was not carried out in advance.”

We agree on the critical role of the calibration curve (here INTCAL20). It seems our novel methodological approach to potential calibration artifacts (see L124-129, Fig. S10) was entirely overlooked by reviewer#1. To avoid the risk of similar misperceptions by other readers, we have conducted additional analysis to extend this central issue in a revised version of our MS. The new analysis confirms the result shown in Fig. S10: SPD-based growth cycles show no relation with artificial effects due to the variable INTCAL20 slope.

We are not aware of the “necessary selection procedure” proposed by the reviewer to restrict to “unaffected time periods”, which would (1) make SPDs obsolete for most of the Holocene and (2) ignores the statistical approaches dealing with varying calibration slopes developed by the UCL group (Crema, Shennan, et al; e.g. *calibrate* function in RCARBON <https://cran.r-project.org/web/packages/rcarbon/vignettes/rcarbon.html>).

“b 14C data from archaeological features date deposited remains of human activity. The deposition frequency of prehistoric societies depends on e.g. economic and ritual practices. If these change, the quantity of deposition also changes. Accordingly, SPDs or RGRs can only be used as indications of relative population trends if the economies of societies or ritual practices do not change significantly. This was not taken into account ...”

Analogue to the calibration issue, this skepticism about the SPD approach fundamentally questions an entire branch of research, which has also been prominently published (e.g., 5 papers in *Nature Communications* and more in other journals of the *Nature* family). Typical study periods of several millennia exceed cultural temporal units of a few centuries. We acknowledge that similar concerns about the SPD approach are widespread among archaeologists, which motivated our extensive cross-validation with independent reconstructions (see below). If the dependence on cultural/socio-economic practices could significantly distort SPDs, the synchronicity among continental growth cycles shown in Fig. 2 appears even more astonishing, as these cultural shifts must have been somehow synchronized worldwide.

“3) As the SPDs and RGRs still reflect (at least partially) the structure of the calibration curve (cp. MC 2a) and the calibration curve in itself mirrors solar activities, the correlations between the solar and the RGR curves might be labelled a circular argument.”

See above. The old and the newly produced evidence clearly speaks against a circularity.

“... Furthermore, 20 examples are cited in Table 4, but 15 of which are also based on the SPD proxy and therefore cannot be used for verification here. In this respect, the correlations found (lines 130-144) are not surprising. In contrast, Results that bring together over 150 studies across Europe that are not based on SPDs and place the archaeological contexts at the centre of the analyses come to completely different conclusions (Müller/Diachenko 2019, Müller 2014, Müller 2015; Schmidt et al 2021).

Other working groups (cp. Shennan,UCL) use palynological sequences...”

None of the 20 independent proxy time-series for changes in population size (Table S4, Fig. S2) is “based on SPDs.” Perhaps Reviewer #1 means that underlying chronologies often

involved C14 dates, as indeed 15 time-series used a variable degree of radiocarbon dating (mixed with, e.g., typochronology). However, as discussed above, radiocarbon dating does not introduce correlations with the SPDs, even if the same calibration curve had been used (which is not the case). Most of the 20 time-series describe site density and thus rely on a very different sample set and methodology than the SPDs; two time-series originate from palynological sequences such as from Feeser, Hinz et al., and one time-series is produced according to the Cologne protocol of Zimmermann et al. Despite their different spatial scales, put together they fit surprisingly well the broad-scale SPD-based trajectory (Fig. 4c). We cannot understand why this unprecedented collection of published (non-SPD) estimates for population shifts across Europe should be entirely disregarded.

We also disagree that our results lead to “completely different conclusions” compared to previous works. The population curve of Schmidt et al. (2021) entails a single data point for our study period, while the reconstructions of population density by Müller (2015) and Müller & Diachenko (2019) mostly have a resolution of 500 years, too coarse to be fully comparable to our results. However, a rather continuous population increase over the Holocene is in line with Fig. S8. Müller (2015) also reported an acceleration of growth at 4.5-3.5kaBP (Southeast Europe) and 4.0-3.5kaBP (Central E), which can be linked to multicentennial regional booms in our reconstruction (Fig. S7). As a single discrepancy, Fig. 5 in Müller (2015) indicated a strong decline in Central E and South Scandinavia at 4.0-3.5kaBP, which has later been reversed by Müller & Diachenko (2019) to a population rise, as now confirmed by our regional results. Furthermore, inherent to the sub-continental reconstructions of Müller & Diachenko (co-author also of our study) is a strong positive growth phase in 8.2-7.2kaBP, while for this period, our high-resolution European RGR shows two strong booms and an intermediate bust. Moderate growth booms around 5.3kaBP as well as a strong decline reported for Southeastern E around 6.2kaBP agree with either our continental RGR or the trajectories for the respective regions in Fig. S7.

Therefore, alike for the other concerns of Reviewer #1, we doubt the thoroughness and accuracy of the underlying argumentation.

Finally, we couldn't reproduce his/her experience that our script and data upload at GITHUB was “not accessible” (perhaps some browsers need an additional “https://”).

In contrast, we find most of the points raised by the other reviewers supportive or helpful for improving the manuscript, while they do not invalidate our major conclusions.

We hope our arguments lead to a re-assessment of the review and would greatly appreciate it if you could reconsider your decision. In that case, we would resubmit a MS revised according to all reviewers' concerns together with a letter explaining ‘point-to-point’ those revisions.

Sincerely,

Kai Wirtz, on behalf of all authors

Nature editorial offices
The Campus
4 Crinan Street
London N1 9XW
United Kingdom

Institute of Coastal Systems
Analysis and Modeling
Ecosystem Modeling

Head of Department
Prof. Dr.
Kai W. Wirtz
T +49 4152 87-1513
M +49 160 75 77 327
kai.wirtz@hereon.de

Rebuttal NCOMMS-24-01345

June, 30 2024

Thank you for handling and especially for reconsidering our manuscript “Multicentennial cycles in continental demography ...” (NCLIM-21050827). We also thank the four reviewers for their elaborated comments. Please find below our ‘point-to-point’ response to all reviewers’ concerns, which also guided the revision of the MS.

In your decision dated May 29, 2024, you referred primarily to Reviewer #1's criticism regarding “summed probability distributions to reconstruct demographic trends and how up-to-date climate discussion is.” As we will show below, Reviewer #1's concerns appear to be based on misperceptions rather than actual weaknesses in our approach. Yet, the comments helped us clarifying certain aspects of our approach.

Comments by reviewer #1

1a) “A major shortcoming of this partially interesting study is the little corroborated correlation between climate stability and population growth rates. What is the hypothesis and rationale behind the oversimplified assumption that varying activity of the sun, if the latter is a main driver of regional climate variability (as stated in line 64 despite of little consensus in the climate community)?”

We do not understand what the reviewer means with “little corroborated correlation”. Is it that the correlation between “climate stability and population growth rates” (RGR) is small? This would contradict the results of our statistical analyses: a trimmed correlation of 0.79 between climate stability and RGR (Fig S4b) is commonly classified as “highly correlated” (see also Fig.S5). Or does the reviewer simply ask for a “rationale” for this high correlation? A mechanistic link between climate stability and subsistence success of both hunter-gatherers and agriculturalists is intuitively clear as discussed in the MS. However, the mechanistic rationale for the high correlation found between RGR and solar activity is indeed less obvious. In the revised MS, we now emphasize the empirical nature of this link, through, e.g., the new title, changed text incl abstract, and the adaptation of Fig.3 (see 3b,f)

1b) “In addition, the reconstruction of the relative demographic trends is not methodologically convincing in all respects and appears to contradict other results in some cases.”

See 1d) below

1c) “Nor the climate discussion nor the climate data base appears state of the art ... This is for instance reflected in the referenced papers where the vast majority is older than 2010. Accordingly, recent studies and break throughs are almost completely missed not only in the discussion, but also in the database. Here recent global compilations are available (e.g. Kaufman et al. 2020) which provide a new baseline for this kind of studies, and which integrate most of the data collection here presented together with a much too long and somehow old fashioned reference list of references.”

Climate proxy compilations select data based on specific purposes. Kaufman et al. (2020) aimed at reconstructing global Holocene temperature and collected proxies with a known relationship to temperature. We targeted climate variability and selected proxy time-series with higher temporal resolution and coverage than most K2020 records. While we do not believe that, in paleoclimatology, the age of publication strongly correlates with data quality, we checked the statistics: the average publication year for the K2020 database for Europe is **1995**, while ours is **2010**. Therefore, we cannot comprehend how “a vast majority [of our records] is older than 2010” or how we are behind “state-of-the-art.” Our compilation is accessible in a popular repository. No rationale is given for why the discussion of solar influences on climate, co-authored by leading experts in the field, isn’t up-to-date.

1d) “Demographic estimates can only partly be estimated using SPDs of large numbers of archaeological radiocarbon date. There are two aspects that make the SPDs appear problematic as a proxy for demographic activities:

In segments of ‘steep slopes’, but also ‘plateaus’ of the calibration curve, the total calibrated values are influenced by the structure of the calibration curve. In this respect, only the unaffected time periods can be considered representative due to these methodological problems. This is also not overcome by producing RGRs because the necessary selection procedure was not carried out in advance.”

We agree on the critical role of the calibration curve (here IntCal20) especially because the IntCal curves in part depend on proxies also used for solar activity. Apparently, our novel methodological approach to potential calibration artifacts (L136-141 and Fig. S10 in old MS) was entirely overlooked by reviewer#1. To avoid the risk of similar misperceptions by other readers, we conducted additional analysis to discuss this issue more prominently in the revised MS. The new analysis confirms the old result: fluctuations due to the variable IntCal20 slope are much smaller than and only sporadically synchronous with SPD-based growth cycles (see new section “Comparison to calibration artifacts” and extended Fig. S10). We are not aware of the “necessary selection procedure” proposed by the reviewer to restrict to “unaffected time periods”, which would (1) make SPDs obsolete for most periods and (2) ignores the statistical approaches addressing the uncertainty arising from varying calibration slopes developed by the UCL group (Crema, Bevan et al: *calibrate* function in RCARBON <https://cran.r-project.org/web/packages/rcarbon/vignettes/rcarbon.html>).

1e) “¹⁴C data from archaeological features date deposited remains of human activity. The deposition frequency of prehistoric societies depends on e.g. economic and ritual practices. If these change, the quantity of deposition also changes. Accordingly, SPDs or RGRs can only be used as indications of relative population trends if the economies of societies or ritual practices do not change significantly. This was not taken into account ...”

Analogue to the calibration issue, this skepticism about the SPD approach fundamentally questions an entire branch of research, which has also been prominently published (e.g., 5 papers in *Nature Communications* and more in other journals of the *Nature* family). Typical study periods of several millennia exceed cultural temporal units of a few centuries. In contrast to all these studies, we had introduced a down-scaling approach resolving smaller and time-variable regions for time windows of 400yr, which are likely connected to individual cultural units (see Methods). This down-scaling approach confirmed the results of the classical (pooled) method as visible in Fig. S9. While possible effects of cultural change on SPDs as hypothesized by Reviewer#1 are for the first time addressed methodologically by our study, no critical effects on results were indicated at high data density. We acknowledge that similar concerns about the SPD approach are widespread among archaeologists, which motivated not only methodological developments but also our extensive cross-validation with independent reconstructions (see below 1h).

1f) “As the SPDs and RGRs still reflect (at least partially) the structure of the calibration curve (cp. MC 2a) and the calibration curve in itself mirrors solar activities, the correlations between the solar and the RGR curves might be labelled a circular argument.”

See above (1d). The old and the newly produced evidence clearly speaks against a circularity.

1g) “The authors are obviously aware of the problem mentioned in MC2b [low accuracy of SPDs]. Accordingly, they argue that the ‘RGR reconstructions would be “well within the rates estimated from independent analyses based on Mid-Holocene house and village structures”. But only two studies are cited in favour of this.”

In the revised MS we expanded the comparison to existing population trajectories **at a much coarser temporal resolution** such as by Zimmermann (2009), Müller (2015) and Müller & Diachenko (2019, the latter co-author also of our study). Almost all reported patterns were found to be congruent with our high-resolution reconstruction (see new section “Confirmation by reported changes in population size”).

1h) “... Furthermore, 20 examples are cited in Table 4, but 15 of which are also based on the SPD proxy and therefore cannot be used for verification here. In this respect, the correlations found (lines 130-144) are not surprising. In contrast, Results that bring together over 150 studies across Europe that are not based on SPDs and place the archaeological contexts at the centre of the analyses come to completely different conclusions (Müller/Diachenko 2019, Müller 2014, Müller 2015; Schmidt et al 2021).

Other working groups (cp. Shennan, UCL) use palynological sequences...”

None of the 20 independent proxy time-series for changes in population size (Table S4, Fig. S2) is “based on SPDs.” Perhaps Reviewer #1 means that underlying chronologies often involved C14 dates, as indeed 15 time-series used a variable degree of radiocarbon dating (mixed with, e.g., typoschronology). However, as discussed above, radiocarbon dating does in general not significantly distort patterns in SPDs/RGR. Most of the 20 time-series describe site density and thus rely on a different sample set and methodology than the SPDs; two time-series originate from palynological sequences such as from Feeser, Hinz et al., and one time-series is produced according to the Cologne protocol of Zimmermann et al. Despite

their different spatial scales, put together they fit surprisingly well the broad-scale SPD-based trajectory (Fig. 4c). We cannot understand why this unprecedented collection of published (non-SPD) estimates for population shifts across Europe should be disregarded.

We also disagree that our results lead to “completely different conclusions” compared to previous works, but actually found the opposite (see above 1g, new section “Confirmation by previously reported changes...” integrating the works of Müller/Diachenko). The population curve of Schmidt et al. (2021) entails a single data point for our study period. Therefore, alike for other concerns of Reviewer #1, we doubt the thoroughness and accuracy of the underlying argumentation.

1i) “A ‘method-independent’ verification of the RGR reconstructions is therefore not provided in the article. Existing studies that used archaeological arguments to support a population reconstruction tend to contradict the results presented.”

See replies above (1d-1g)

1j) “The correlation between “solar activity”, climate, subsistence and population density is reduced to a broadening perspective. The reconstructed dependency of population density from solar activity is due to a generalisation of data in a reduction of accuracy.”

We are not aware of any study in paleo-demography and paleo-climatology that encompasses roughly four spatial scales from sub-regional to regional, continental, up to global scale. By contrast, most archaeological studies concentrate on local to sub-regional trends. We agree that some of our results such as synchrony of human growth rates with solar activity would not be visible at these local to regional scales (for example, see Fig. 3A-B in Bevan et al 2017), but this does not contradict their emergence at larger spatial units. We cannot extract from the reviewer’s comment where our statistical analyses of an unprecedented amount of data might have lost accuracy.

1k) “In respect to climate reconstructions the manuscript is not state of the art. In the case of archaeological population reconstructions, those that are not based on SPDs are largely ignored. The compiled data are obviously not compiled by the authors.”

See replies above (1c,1g). Our radiocarbon data-set for Europe is unique, its sources well described (e.g., Methods, Tab S1, GITHUB repository) so that we cannot comprehend how “data are obviously not compiled by the authors”.

1l) “As the reconstruction of population densities is only based on 14C-data, additional evidence is needed. Methodologically independent further evidence has to be integrated and discussed.”

See above (1g-1h)

1m) “The selected climate records have to be added by already existing new compilations.”

See above (1c)

1n) “From the supplements it is unclear which 14C-data-sets have been used for which calculation. The reference to general data compilations (e.g. RADON) or references does not provide the reader with the possibility to identify the data which were used. May be,

they are in DOI [10.5281/zenodo.10467363](https://doi.org/10.5281/zenodo.10467363) or github.com:kaiwirtz3/holocene (unfortunately not accessible).”

We couldn't reproduce his/her experience that our script and data upload at GITHUB was “not accessible”. Perhaps some browsers need an additional “<https://>” which is now added to the GITHUB link. The upload included files containing all radiocarbon data before and after merging the p3k14c data-base.

1o) “Tab. 1: RADON lacks <https://radon.ufg.uni-kiel.de/>; RADONb lacks Kneisel et al. 2013” Link added; the RADONb web-page suggests Rinne, Kneisel, Hinz, et al (2024) as reference, which is added as well.

Comments by reviewers #2 + #4

2a) “Beginning on line 170, for example, the authors claim that “Against expectation, RGR of Northern Ireland often fluctuated in antiphase to tree ring width itself... which suggests that conditions beneficial or detrimental to tree growth could have had the opposite effect on human population growth.” Here, the authors should consult B. Campbell and F. Ludlow, (2020). Campbell and Ludlow indeed note that conditions in this region favorable to tree growth are often inverse to those favorable for crops. This helps to explain the authors' finding, which is not at all “against expectation” to regional experts. The authors should therefore tweak their wording, but it is significant to us that they accurately and independently made this rather obscure finding.”

We thank the reviewers for pointing to the work of Campbell and Ludlow (CL2020). However, CL2020 covers a different temporal scale, thus interannual and often season specific variations of climate in Ireland within two centuries. Our study addresses decadal-to-centennial variations over the entire Holocene, also using a different dendrochronological data-set (provided by the same scholar, Mike Baillie). CL2020 contains both evidence for and against a negative relationship between tree ring width and subsistence success, which motivated a rewording in the new MS.

2b) “More broadly, we note that some historians have long argued that shifts in climate stability have influenced human history at least as much as changes in climate state. In 1980 Jan de Vries, for example, ... in 2018 Dagomar Degroot made similar claims ... The scale considered by these historians is far smaller in time and space than that analyzed in this manuscript. Yet to us it does seem telling that processes on a relatively small scale seem to confirm the claims made in this manuscript that concern a much larger scale. We would suggest that the authors briefly acknowledge this apparent confirmation.”

Again, we are grateful for these valuable hints. We extended the climate-society discussion citing deVries 1980 and the review article by Degroot et al 2021 *Nature* (L212-214).

2c) “The authors do well to evaluate the strengths and limitations of the paleoclimatic reconstructions they employ, though we think that not all of the proxies examined in this manuscript can be characterized as “high resolution” (line 70).”

The term “high resolution” needs to be understood in relation to the length of the study

period (here, 6 millennia). We replaced “high resolution” with “decadal-to-centennial”.

2d) “We do think, however, that this manuscript is less accessibly written than many publications in Nature Comms. Not only are the manuscript’s descriptions of complex, correlated changes unnecessarily wordy in places, but the manuscript’s argument is not expressed with sufficient clarity in its first two pages. Instead, the argument unwinds slowly as the authors introduce data and acronyms. We suggest that the authors succinctly provide their argument on the first or second pages of their manuscript...”

We further shortened sentences where possible and moved parts of the Introduction to the latter text to streamline the argumentation in the initial part of the MS.

Comments by reviewer #3

3a) “in the paper the use of boom/bust scenarios (e.g. line 46 but also the rest of the paper) does not follow convention and therefore introduces significant confusion in the authors' interpretations. RGRs are estimated as relative growth rates with $[RGR = \{SPD(t+\Delta t) - SPD(t)\} / \{SPD(t)\Delta t}]$ - a calculation discretely tucked away in the methods section. There is precedent in the use of growth rates to analyze 14C demographic proxies (Arroyo-Kalin & Riris 2021, RTSB). But, growth rates do NOT IDENTIFY booms and busts. They are NOT EQUIVALENT to Boom or Busts either. This is shown by the authors themselves in Figure S.8. In most previously published SCPD studies, a boom is defined as overshoot above the expected demographic growth model; a bust is falling below the floor of the expected demographic growth model. RGRs are neither, as shown in Fig. S8. Each RGR upturn or downturn is an acceleration or de-acceleration of growth rates, nothing else until it is shown that it expresses a time-series that exceeds (+ or) the expected demographic model.”

We agree with the reviewer that booms and busts (b&b) are frequently defined based on overshoots and depressions in the SPDs, which is facilitated by the default output of the popular RCARBON package where these overshoots and troughs are marked by color. By contrast, the perhaps first **causal** analysis of SPD-based paleo-demography by Lima et al 2023 *Phil.Trans.R.Soc.B* used RGR as central target variable. The RGR-based approach connects to **all mechanistic** population models being founded on the functional dependence of RGR (and not of population density) on internal or external factors.

We think that the accustomed SPD-based usage of b&b can even obscure causality patterns. For example, Bevan et al (2017) related paleo-demography for the British Isles with solar activity (TSI) and identified three “bust” periods that overlap with minima in TSI (see their Fig.3A-B). However, these minima in TSI are located rather in the late “bust” phase and coincide with uplifts in SPDs (positive RGR; indicating subsistence success). Therefore, the data presented by Bevan et al turns their own conclusion of coinciding “population **down**-turn with ... reduced solar activity” into the opposite, i.e. low TSI during population **up**-turns; the latter confirms the results of our study (Fig.2e-j). We added this argument to the Discussion (L272-277) and now better explain our growth-based definition in the Introduction (L39-44). Later in the MS, b&b are often attributed to “growth” for better distinction from the accustomed SPD-based notion.

3b) “Co-patterning with solar activity is significant. That the latter dictates the condition for

population boom and busts is not demonstrated by the paper.”

Cycles in continental growth rates run often synchronous with cycles in (negative) solar activity. Our previous presentation of this outcome may have been misleading as we can so far only speculate about the mechanistic origin of this correlation (see Text S1). We have hence adjusted Fig.3 as well as the entire text including the abstract and the title so that the solar link appears less prominently in the revised MS. However, in doing so we avoid invalidating the identified synchrony of cycles in solar activity and human growth but wish to motivate the investigation of mechanistic origins of the links between solar activity, climate stability, and subsistence success in future studies (see, e.g. L277-278, 3f)

3c) “The section S1 “Solar influence on climate fluctuations” is too important for the overall reasoning presented in the paper to be buried in SM. A summary of its key points deserves to be part of the introduction of the paper or be included in the section “Synchrony with solar forcing”. Moreover, S1 ends with an insufficiently substantiated statement that verges on hasty generalization (406): “Based on this finding, we here hypothesize that lower solar activity may on a short timescale enhance the likelihood of weather extremes but on a long timescale reduce variations in the mean climate state”. This hypothesis is never tested in the paper. I can see how rapid downturns in solar activity may have short-term effects on weather extremes (e.g. through atmospheric blocking patterns, as suggested by authors) but their influence on the long-term mean climate state is less clear: a compounded effect over the long-term is likely an outcome of rhythm and frequency of rapid downturns, not a long-term property per se.”

Section S1 was slightly extended but remains a short review on what is currently known about the “Solar influence on climate fluctuations”. S1 is summarized in the revised Discussion (L290-296) and concludes with a hypothesis for processes behind the **empirical** relation of RGR and partially also climate stability on TSI. The reviewer is entirely right that we cannot substantiate this hypothesis. We now better clarified the complications of a full mechanistic underpinning, which thus is beyond the scope of our study. This restriction on the empirical nature of the solar link is now emphasized by, e.g., the new title, changed abstract, the adaptation of Fig.3 and Discussion (e.g., L277-278, L290-296). See also 1a, 3b,f.

3d) “Line 39 - If comparisons of continental datasets appear to be poorly synchronised (paragraph 1), how would focussing on a single region be a legitimate way to generalise to “validate any global inferences ... at continental scales” - another hasty generalization?”

First, we now have better emphasized that regional trajectories reported in previous studies were poorly synchronized **within** the individual continents (L28-30). The sentence did not say anything about the synchrony between continents. In fact, we found a relatively high synchronization between continental RGRs. However, the reviewer is right that an independent validation for a single continent cannot be easily generalized to the global reconstruction. We thus omitted “global” in L33.

3e) “Lines 70, 71: both statistical adjusting of chronologies to improve synchrony and the ways in which time-series were aggregated into principal components are difficult to follow in SM.”

We extended the description of how we treated paleoclimate proxies in the Methods.

3f) "Lines 78-83. Models IV, 2V, 3V - all unclearly explained in the paper. Particular, Fig. 3 (left) is difficult to make sense of. In S1 it is suggested that lower solar forcing would impact short term climate by inducing more extreme conditions (this made sense to me). Fig 3, Left would seem suggests the opposite (high solar activity-> variability-> failure, light blue arrows), and conversely? Can this be improved for the sake of communicating more effectively. In addition, pathway 2V should be described in words (high solar activity leads to climate variability which leads to climate failure) and it should be reconciled with discussion of short-term versus long-term effects (S1, line 406) reconciled"

We have rewritten the introduction of the logit model (now L77-84). Furthermore, we changed Fig.3-left by (1) streamlining the design and removing confusing details, (2) adding a box for linking the variables to the input configurations '1V'-'3V' of the logit model, and (3) expressing the yet uncertain mechanistic underpinning of the link between solar activity and climate stability.

3g) "Lines 88-90: This highest spectral agreement with the '3V' logit regression model is hard to understand."

Now clarified

3h) "Line 97: These three population "collapses" cannot be seen as such on the basis of RGRs. There's no collapse, just de-acceleration.

Here we disagree. A negative RGR is not "just de-acceleration" but reflects a down-turn in a demographic proxy. A down-turn with RGR = - 0.2% per year means that in average 20% less activity is found in a cultural horizon dated one century later. Yet, we have replaced "population collapses" by "busts". "De-acceleration" is instead described by a negative second order derivative, thus a negative change rate of RGR.

3i) "Shennan et al. 2013 show clearly that a "proper" bust in one European region is not reflected in another (see their Figure 3,...French German regions for the ~6.4 BP chron)."

As already stated in the text (L112-113), our regional results in general confirm the reported heterogeneity between regional dynamics but also feature sporadic large-scale synchronization (L113-116). This synchronization was not visible in previous studies because of their much smaller data sets and incomplete spatial coverage. For example, at 6.4kaBP our regional reconstruction in Fig.S7 displays heterogenous RGR in central and western Europe, but widespread negative RGR in Eastern Europe, which in total sums up to a continental (growth) bust.

3j) "The differential impact of arrival of Neolithic agrarian societies across the European region is, furthermore, not considered (Bevan et al. 2017, PNAS)"

Good point. We already discussed the migration of agro-pastoralists into and through Europe in some detail in S2 and added a statement in the Discussion that inter-regional migration does not affect the conclusions especially on the continental scale - since losses in the origin region are compensated by gains in the destination region (L265-267). We feel that a more in-depth discussion of the growing archaeogenetic literature or of regional dynamics is beyond the scope of this MS.

3k) “Figure S7 hardly supports a population bust at c. 4000 BP”

We thank the reviewer for her/his accurate attention. Indeed, the 4kaBP bust is the single event of the averaged “region based” RGR not inherent to the “pooled” RGR (see Fig.S9, perhaps the 8.4 bust could be regarded as 2nd example). The sharp edge of the “region based” trajectory at 4kaBP indicates that a very negative RGR for a short period (<50a) is responsible for the low value. The anomaly occurred in parts of southern UK (see also Fig. S7 - though time-averaged values are shown). We do not see a reason to exclude a regional growth rate of about -1% per year (that may also describe outmigration, see Bevan et al 2017, or 3j), but have indicated the uncertainty of this bust in the revised text (L122-124) and removed its explicit citation in L99.

3l) “Line 104: Can the authors check the underlying for Fig. S6 ? Time shifting the 'Climate stability' curve by about 1000 years seems to provide a much better match between stability and solar forcing, which would agree much better with the suggestion that short-term lower solar activity directly impacts climate stability.”

Yes, the reviewer observed the data well: shifting one of the two time-series can increase the synchrony. For example, moving TSI forward in time by 200a, trim correlation (tr) gets very high ($tr=0.88$). Assuming TSI as causative for climate stability, we need to shift TSI backward in time. In that case, only after a lag of 800a we get $tr=0.71$ and much smaller or negative tr for other lag times. However, it is difficult to explain such a lag time of nearly one millennium between solar activity and climate stability so that we do not see how we can gain understanding of the mechanism underlying the solar link, which is described differently in the revised MS (see 3b,c,f).

3m) “Line 239: Most of the time, and especially with RGRs as opposed to Boom / Bust scenarios, what is more clearly tracked is de-acceleration in birth rates rather than large scale factors acting rapidly. RGRs do not provide any information about population decimation on their own.”

As already outlined above (3a+h), RGRs have no relation with “de-acceleration in birth rates”, but are central target variables of a causality oriented analysis.

While we found many points raised by the reviewers supportive or helpful for improving the manuscript, these do not invalidate our major conclusions.

We hope our arguments lead to a re-assessment of the review and would greatly appreciate if you could reconsider your decision.

Sincerely,

Kai Wirtz, on behalf of all authors

Dear editors and reviewers,

Please see below the 'point-to-point' response to all critical concerns raised during the review of our manuscript "Multicentennial cycles in continental demography ..." (NCLIM-21050827). The concerns and responses also guided the revision of the MS.

We very much thank the old and new reviewers for their helpful comments.

Comments by reviewer #2/4

2a) "The author(s) have responded to our suggestions, implementing them skillfully or else rejecting them with an explanation that satisfies us. They have also revised their manuscript so it is more accessible than it was.

We recommend citing Degroot 2018 ("The Frigid Golden Age") rather than Degroot et al. 2021 ("Towards a Rigorous Understanding) for the point on climatic variability being more important than trends in some historical work.

Done, but we inserted Degroot et al. 2021 elsewhere.

Comments by reviewers #5 were all positive and are therefore not listed here.

Comments by reviewer #6

6a) This paper is part of a growing literature of comparative human population ecology during the Holocene. Yet, the author does not acknowledge nor, more importantly, really build on this literature to illustrate how this study advances our knowledge of long-term human population dynamics. This has two consequences. (a) It stagnates the critical discussion of ideas. (b) I believe that this contributes to the lack of a coherent theoretical framework within which to ground the results of the current paper.

In a substantial revision of Introduction and Discussion, we have re-build the storyline and set-up a clear-cut research question, i.e. the relative importance of endogenous versus exogenous drivers of human paleodemography. This implies that we do not follow the strict deductive approach proposed by the reviewer, though we now partially refer to some papers later indicated by him. As also explained below in (6d), our results do not support simple underlying theories. For example, Morans' theorem can help us to link the interplay of endogenous and exogenous factors found in our study to analogous situations in ecology, but it does not predict important features of the reconstructed pattern such as the dependence on spatial scale, or the degree and type of external control (climate stability instead of state). Therefore, we believe that "a theoretical framework for long-term human population dynamics" should integrate a larger set of mechanisms than is typically formulated by, e.g., the resilience literature. Some important aspects of this more integrative concept are visualized in Figure 3. We have furthermore added a link to a foundational integrated modeling work (Wirtz & Lemmen *ClimDyn* 2003 - a paper ignored by the recent modeling literature provided by the reviewer). There, long-term human population dynamics in the Holocene has been formalized and simulated for all world regions in terms of a combination of processes such as technological shifts, adaptation in diverse subsistence styles, migration, or climatic fluctuations. This work also proposed a functional relation between population growth and Net Primary Productivity, as later done by Freeman et al JARCS 2020 or – in a reduced way– by Freeman et al PNAS 2024.

(6b) The current manuscript claims that this study is almost completely new in scope and topic.

And this is a disappointing strategic decision by the author because it stagnates the critical discussion of ideas that is central to knowledge growth in science. For example, Freeman et al. 2018 study the synchrony of radiocarbon SPDs with each other and with solar energy at a global scale. The current paper appears to build on the conceptual framework laid out in the Freeman et al 2018 paper. For instance, Freeman et al. state:

“The synchrony of energy consumption among human societies, at a global scale, could result from two global mechanisms. First, human societies may all respond similarly to fluctuations in an external driver—the so-called Moran effect (18).....Thus, we might expect that fluxes in solar energy cause human populations to synchronize, and, if so, human populations in different biophysical environments should synchronize with each other and with the influx of solar energy....Second, direct interactions such as trade and migration, as well as indirect interactions (e.g., common disease vectors or indirect trade), may cause the synchrony of energy consumption among human populations. (p. 9963).”

In sum, either external or internal drivers may cause human populations to synchronize. The current paper clearly follows on this conceptual investigation and advances on the earlier study by analyzing a lot more data. ... Interestingly, the current paper and the 2018 paper come to different conclusions, and it is important to understand why they come to different conclusions. It would be really exciting and interesting if solar forcing were driving synchrony between human populations.

I suggest that the author re-frame the paper as an advance on earlier studies and the growing literature engaging in global comparisons of long-term human population dynamics. Honestly discussing these earlier works and how this study advances on them is a more powerful framing than claiming to be the first to study such issues.

We are grateful to the reviewer for pointing us to the work of Freeman et al 2018. A probable cause for why this paper had been under our radar is that it uses “energy consumption” instead of “population density” as major variable. And, yes, the identified global synchrony pattern makes this paper an important point of reference, as now acknowledged very prominently in the re-framed Introduction and Discussion. Also, Freeman et al 2018 proposed a similar test for calibration artefacts (see renewed SI-section “Comparison to calibration artifacts”). It is especially interesting that Freeman et al studied the role of solar forcing but arrived to an opposing conclusion w.r.t. to our result. The possible reasons are summarized in the new Discussion and here explained in more detail: Freeman et al (1) relied on a much smaller dataset, (2) conducted their analysis at a smaller scale, used (3) SPDs instead of RGRs, and (4) a different metrics.

(1) Different data can obviously lead to different results. In addition to the much smaller compilations of C14 dates, Freeman et al refer to sunspot number from Solanki et al 2004, while we use Total Solar Irradiance (TSI) from Steinhilber et al 2012. Our analysis also extends to a wealth of paleoclimate information.

(2) We averaged C14-based population growth (RGR) for a series of spatial boxes centered at Northern Ireland and found a substantial influence of the box size on the resulting phase overlap of RGR with TSI (see new Fig. S9) Similar patterns can be obtained for other center locations. Therefore, at a local-to-region scale (used by Freeman et al) the correlation of RGR with solar forcing might be weak or absent, while it best emerges at a continental scale (or slightly below). This scale dependent transition may reflect both peculiar endogenous dynamics in different regions, but also region-specific changes in climate stability as already discussed for Northern Ireland in the MS.

(3) In all consistent theoretical frameworks known to us, environmental factors do not act directly on population density, but – if at all- on population growth rate. We re-emphasized this aspect in our revision. The choice by Freeman et al of using SPDs instead of RGRs introduces a time-lag of about half of the dominant period (see Fig.3), which may blur synchronization between climatic forcing and population responses.

(4) Freeman et al discretize their time-series into a sequence of symbols and then apply a similarity metrics from information theory (“mutual information”). Due to discretization, small excursions are treated equal to large ones so that amplitude information is almost lost. The discretization effect may not only amplify noise but also calibration artefacts. The latter were also discussed by Freeman et al themselves and are in our work assessed as multi-centennial perturbation with small amplitude (Fig.S17), which may distort mutual information. See also below (6h)

(6c) A few papers that engage in comparative studies of archaeological radiocarbon. Freeman2024, Riris2024, Bird2020, Freeman2018, Jorgensen2022

We don’t understand what “comparative” means in this context. We already cited abundant literature on archaeological radiocarbon and now have included Freeman2018, Riris2024, and Bird2020 as these directly relate to the renewed Discussion.

Freeman2024 offers a delay mechanism -probably describing soil degradation and forest re-growth?- as mathematically easy solution to the problem of producing oscillatory dynamics, which however is difficult to compare with empirical evidence.

The “synchrony” suggested by Jorgensen2022 is limited to only two regions and a very rough accuracy, and thus not very compelling.

(6d) One of the main weaknesses of the paper is that it lacks an overall theoretical structure. This contributes to the paper not reading very well, as noted by previous reviewers. I think that engaging with some of the previous studies of global scale comparisons noted above can help the author develop a better narrative structure that critiques and builds on previous work. For example, in the current version of the manuscript, the author launches into a discussion of the importance of climate stability, but this comes off as a purely inductive exercise in which the author is searching for associations, and then post-hoc justifying why they are meaningful. There is good theory in the resilience literature that would suggest that climate uncertainty is much more difficult for humans to deal with than predictable extremes. This is because humans use rules and infrastructure (culture) to create flows of resources, but inevitably the rules and infrastructure must be designed for a given climate regime. If climate uncertainty increases, this will stress existing infrastructure systems and social rules for allocating resources.

We sought to make our storyline more appealing by, e.g., better defining the research question (see above 6a-b). While now citing various pieces of the resilience literature, we are reluctant to dive more deeply into this branch of research since most resilience studies make use of simple models describing a single mechanism, which according to our results is a highly improbable assumption. Our deliberate restriction to statistical modeling provides only first indications for causal relationships, which need to be better assessed by future empirical studies or by integrated mechanistic modeling (see 6a). Yet, rethinking our results from the perspectives elaborated in the resilience literature helped us to sharpen our interpretation, e.g., by considering *manageable time scales* in the Discussion.

(6e) In general, the author has effectively commented on previous reviewer critiques. However, a critical part of the methodology of this paper is not well presented and requires revision for publication. The authors do a good job of describing how they constructed the SPDs and RGR. They also describe how they reconstruct paleoclimate stability fairly well.

The major short coming is that the authors do not describe their methods for assessing synchrony nor identifying the dominant modes. I tried to figure it out from the r and matlab code, but these files are unorganized and not commented very well, thus, I could not figure it out in a reasonable amount of time.

We have extended the Methods in particular w.r.t. the description of our two synchrony measures and of the spectral analysis (the Fourier Analysis is very common in engineering and physical

sciences but not so much in social and biological sciences). Moreover, we have improved accessibility of our script code, which is now updated in the GITHUB repository.

(6f) So how do you calculate phase overlap? It seems from Fig. 1 that this is done by converting the time series using some kind of transform (Fourier transform??) as assessing the degree of locking (as on a unit circle).

We couldn't figure out which element in Figure 2 could lead to misunderstandings (Fig. 1 just shows maps of the study domains). The treatment of the time-series has been described in the Methods and should also be clear from the caption ("All time-series ... were normalized (division by standard deviation σ), smoothed and detrended"), thus there is no Fourier transformation involved.

(6g) This is really important to make clear because using a method that assumes that the underlying variables are continuous and not noisy will artificially inflate the degree of phase locking. I have no way of assessing the robustness of the current results to the choice of method. I can at least say that the hd Pearson coefficient, even in the de trended data, is probably not the best measure of association.

The treatment of the time-series (normalization, smoothing, and detrending, see above) are standard procedures in time-series analysis, which help to focus on the scale of interest. Smoothing removes high frequencies (noise) and detrending filters out low frequencies that may be of importance in studies on longer-term trends, whereas our analysis concentrated on (multi)centennial variability.

We would appreciate a rationale for why the trimmed Pearson coefficient should be unsuitable. It should not be confounded with an ordinary Pearson coefficient, and shares a similarity with the Mutual Information (MI) used by Freeman et al 2018 as it introduces a threshold in the processing of the time-series (see also a now cited overview description by Alonso et al 2024). While the MI discretized according to whether a values are above or below thresholds, the trimmed coefficient only measures the extremal part, thus filters out low amplitude noise (see 6b).

(6h) In fact, the original title of the 2018 paper noted above was "Solar energy drives human population synchrony." However, after a deep review of how to assess the synchrony of noisy and discrete data, the synchrony between the SPDS and solar energy weakened considerably across scales. The SPDs are discrete and noisy data, no matter how much they are smoothed. I can feel a response coming here, so let me just say this:

At minimum, the paper would improve significantly if the author presents their methods for calculating the phase overlap of the time-series. They should discuss why their method is more appropriate than say, mutual information with a Markov process to construct simulated time-series. Even better would be to compare the methods that they use and the methods that we describe in our 2018 paper. If the results are robust to changing method, then great! If not, then the discussion of the appropriate methods becomes really important to have in the literature so that other researchers can assess and build better methods.

As explained in (6b), the approach of Freeman et al 2018 differs from ours in various aspects (scale, SPD \leftrightarrow RGR, solar proxy, synchrony metrics). We improved the documentation of our methods in the main text and within our script repository (see 6e). We then invested another considerable effort to reconstruct a complicated because now outdated python-environment in order to run the GITHUB scripts provided by Freeman et al. Using mutual information (MI) as synchrony metrics largely confirms the results obtained with our two metrics (phase overlap and trim correlation) as most of the synchronous pairs identified in our study also reveal high MI (see figures below). Examples for high synchrony at different time resolution (100a and 160a) comprise a series of pairs with RGR of Europe (EU) and (i) total solar irradiance (TSI), (ii) climate stability (CS), (iii) shifts in independent occupation data (occ), (iv) RGR of South America (SAM),

and (v) of North America (NA_m), while direct climate forcing seems to be rather unrelated (PCA-1 and PCA-2 of 98 paleoclimate proxies for Europe). At 160a resolution, intercontinental links of RGR gain relevance such as between SA_m and East Asia (EAs) or NA_m and Africa (Afr), or most continental RGRs vary in synchrony with TSI, which together well agrees with the pattern shown in our Fig.2.

MI between EU and TSI exceeds 0.3 not only at intermediate resolution (100-160a) but also at high (50a) or low temporal resolution (200a, both not shown) so that this synchrony can be regarded **robust against changes in scale**. Furthermore, the solar proxy used by Freeman et al 2018 (sun spot number, SSN) reveals **high** MI with NA_m (their core area) at 160a in contrast to the MI shown in their SI-Fig.7. Both differences are probably due to our choice of continental spatial units and of growth rate (RGR) instead of standing stock (SPD). We have added additional material displaying the scale effect in the new SI, and discussed it in the main text.

However, we decided not to include MI as third metrics into our study because of the large information loss inferred from coarse grained discretization (see 6b,g) and of lengthy methodological discussions, which would further complicate and extend both the main text and the SI. We fully agree with the reviewer that assessments of synchrony metrics for uncertain time-series might become an exciting topic of future studies.

Minor Comments/Reading notes:

1. Abstract: "A possible link from multicentennial variations in solar activity to climate stability as a critical condition for human subsistence success, could explain the occurrence and partial phase synchrony of growth cycles worldwide." What is meant by subsistence success here? I don't understand this sentence and general. Please clarify.

Sentence now simplified

2. line 24 "However, going back into the past and focusing on local to regional scales, the archaeological record discloses a more disruptive picture with a series of rises and falls of societies connected to concomitant shifts in population density." This is a statement of fact not backed up by any evidence. Add citations to works that demonstrate this fact. Perhaps describe an example of the general pattern that you have in mind.

This sentence has been deleted.

3. "but human demography and climatic drivers seem to lack spatio-temporal coherence at larger spatial scales." Stated as fact with no evidence to back the claim. Add citations to works that demonstrate this

The paragraph has been restructured. The corresponding "leftover" sentence ("However,

contradicting”) contains a series of citations.

4. line 130 “` `The causality of prehistoric demographic change also remains obscure because very few studies have tackled the problem at the continental scale, and none assessed the climate–society relationship systematically at regional scales.”

This is hard to follow. I am aware of many studies that tackle the problem at a continental scale and assess the climate--society relationship systematically at regional scales. In addition to the Lima et al. 2024 paper cited, here are just a few examples: Riris2024, Bird2020, Freeman2018, Jorgensen2022.

We refined the wording of the statement. Riris et al 2024 aggregated their data at the continental scale, but did not compare with paleoclimate data. Freeman2018 used a single solar proxy with unknown relation to climate dynamics (see our SI chapter S2), while Jorgensen2022 – alike many similar studies comparing SPDs with climate proxies such as Warden et al 2017, Palmisano et al 2021, or Riris & Arroyo-Kalin 2019 use only a few paleoproxy time-series and assessed broad-brushed synchrony “by eye” (see 6c). Also, alike

the large majority of studies they related excursions in possible environmental drivers with peaks and troughs in SPDs (only exception is Bird et al 2020). However, as argued already in the MS, mentioned in the responses to (6b) and (6h) and to a previous review, this approach fails to accurately assess synchronies because of the lag between a causative change in RGR and its effect (changed SPD) as visualized in Fig.3.

5. Line 40 “` `Our usage of RGR allows a more precise causal inference analysis than referring to, e.g., extrema in SPDs because drivers act on growth rate rather than on population density. Analogously, ‘booms’ and ‘busts’ in archaeology are regularly defined based on peaks and troughs in population size, so that the second half of, e.g., a “boom” necessarily features negative RGR. However, a high but declining population size more likely describes a crisis situation than a low but rising population. Therefore, we here distinguish (growth) booms and busts according to the sign of RGR.”

This just seems like an odd place for this text. As a reader, I am still trying to understand why this study is important and what it will report, and then I am thrown into a detailed methodological discussion. I suggest moving this text to the methods section.

Again, many thanks for helping us to improve our storyline. We moved this argument, while keeping it in the revised Introduction. Highlighting the role of RGR for causal analysis may appear trivial for biological/ecological modellers but still is critical in archaeology, sociology, or even human ecology as evident in many responses above.

6. “` `Apart of the period 6.5 to 5.5 ka BP, the demographic trajectories of Europe and South America show a high level of agreement.” Check—should it be “` `Apart from???”

Now corrected to “apart from”.

7. 160: “If the calibrated probability distribution were normalized, trim correlation between reconstructed SPD-based RGR and changes in occupation density would decrease from nearly one to zero (Fig. S11b, Fig. S5), which approves our methodological choice of not normalizing during calibration.” The logic here is difficult to follow. Move all discussion of normalized SPDs to the SI. It would seem that your choice to normalize or not normalize the SPDs affects the results. This does not “` `approve” the non-normalization, it is just a fact. The results are not robust to changes in method (normal vs non-normalized). You need to make an independent argument that non-normalized are more appropriate than normalized SPDs (but please make the argument in the methods and SI rather than here).

Good point, which helped us to create a better text flow both in the main text and the Methods. As stated later in the Methods, the “independent argument that non-normalized are more

appropriate” comes from observations of artefacts due to normalization made in preceding studies (e.g., Weninger et al 2015).

8. `` Our analysis indicates that any climatic shift regardless of the direction could induce stress to human subsistence and growth. ... with previous hypotheses on the resilience of historic societies against climate extremes, but their more subtle susceptibility for long-term climate change [32, 33].”

Here, the reviewer seemed to have forgotten to write down his comment. In the rephrasing of the storyline, we have extended this sentence, also by adding references from the “resilience theory” given in comments.

9. Line 221 ``no or” typo.
changed to “absent or”

10. I cannot access the data and code to review via the links. Both links below appear broken. I had to search for kaiwartz3 on Github and click on the user profile to find the data and code. The code is not really commented at all, thus it is very difficult to follow and replicate the author’s procedures.

We have refactored the code, thereby providing additional comments (see 6e). Yet, it is not true that the code was “not really commented at all”, since in the older version it had at least the density of comments as found, e.g., in the Github code of the reviewer (apart from the preambles). We have up-dated and double-checked the Github and Zenodo links, now provided using latex “`\url{}`”.

Remarks on code availability:

The links provided in the paper do not work. I had to search the user name on GitHub. The code is not well organized and minimally commented. This will make replication difficult?

See reply above

We hope that our responses and revisions can satisfy the concerns raised in the last review round.